# Evaluation of $SO_2$, $SO_4^{2-}$ and an updated $SO_2$ dry deposition parameterization in UKESM1

Catherine Hardacre[1], Jane P. Mulcahy[1], Richard Pope[2,3], Colin G. Jones[3], Steven T. Rumbold[3], Can Li[4,5], Colin Johnson[3], and Steven T. Turnock[1,6]

[1]Met Office, Exeter, EX1 3PB, UK
[2]School Earth and Environment, University of Leeds, Leeds, UK
[3]National Centre for Atmospheric Science, University of Leeds, Leeds, UK
[4]NASA Goddard Space Flight Center, Greenbelt, MD 20771, USA
[5]Earth System Science Interdisciplinary Center, University of Maryland, College Park, MD 20742, USA
[6]University of Leeds Met Office Strategic (LUMOS) Research Group, School of Earth and Environment, University of Leeds, UK

**Correspondence:** Catherine Hardacre (catherine.hardacre@metoffice.gov.uk)

**Abstract.** In this study we evaluate simulated surface $SO_2$ and sulphate ($SO_4^{2-}$) concentrations from the United Kingdom Earth System Model (UKESM1) against observations from ground based measurement networks in the USA and Europe for the period 1987 to 2014. We find that UKESM1 captures the historical trend for decreasing concentrations of atmospheric $SO_2$ and $SO_4^{2-}$ in both Europe and the USA over the period 1987 to 2014. However, in the polluted regions of the eastern

USA and Europe, UKESM1 over-predicts surface $SO_2$ concentrations by a factor of 3, while under-predicting surface $SO_4^{2-}$ concentrations by 25-35%. In the cleaner western USA, the model over-predicts both surface $SO_2$ and $SO_4^{2-}$ concentrations by a factor of 12 and 1.5 respectively. We find that UKESM1's bias in surface $SO_2$ and $SO_4^{2-}$ concentrations is variable according to region and season. We also evaluate UKESM1 against total column $SO_2$ from the Ozone Monitoring Instrument (OMI) using an updated data product. This comparison provides information about the model's global performance, finding

that UKESM1 over predicts total column $SO_2$ over much of the globe, including the large source regions of India, China, the USA and Europe as well as over outflow regions. Finally, we assess the impact of a more realistic treatment of the model's $SO_2$ dry deposition parameterization. This change increases $SO_2$ dry deposition to the land and ocean surfaces, thus reducing the atmospheric loading of $SO_2$ and $SO_4^{2-}$. In comparison with the ground-based and satellite observations, we find that the modified parameterization reduces the model's over prediction of surface $SO_2$ concentrations and total column $SO_2$. Relative

to the ground-based observations the simulated surface $SO_4^{2-}$ concentrations are also reduced, while the simulated $SO_2$ dry deposition fluxes increase.

## 1  Introduction

Anthropogenic sulphur dioxide ($SO_2$) emissions have been the main driver of the historical aerosol effective radiative forcing (ERF) since the mid 20th century (Boucher et al., 2013). $SO_2$ is emitted into the atmosphere from a number of anthropogenic

(and natural) sources and once in the atmosphere $SO_2$ can be oxidised to form sulphate ($SO_4^{2-}$) aerosol, which plays a key role

in both acid deposition, atmospheric aerosol loading and cloud properties, thereby directly influencing the Earth's radiative balance. For Earth system models (ESMs) to have a good representation of the historical climate and thereby give us confidence in their future projections, it is extremely important that they can capture the sulphur cycle. The UK's Earth system model (UKESM1), in common with other ESMs, has a cold bias in the mid 20th century which looks to be associated with an

excessively negative aerosol ERF (Sellar et al., 2019; Seland et al., 2020). A key component of the analysis and development of UKESM1 focuses on the model's sulphur cycle and its link to historical aerosol forcing. Mulcahy et al. (2020) conducted an in-depth evaluation of the aerosol species in UKESM1 and its physical model component, HadGEM3-GC3.1, including $SO_4^{2-}$ and uncovered some interesting differences in the sulphur budget between these two models including differences in the $SO_2$ lifetimes and oxidant loading. We aim to extend their work by conducting a detailed evaluation of $SO_2$ and by probing deeper

into the process level uncertainty of the sulphur cycle.

Sources of $SO_2$ include industry, energy, land-based transport, shipping, volcanoes, biomass burning and marine di-methyl sulphide (DMS) (Feng et al., 2020; Fioletov et al., 2016; Liu et al., 2018; Janssens-Maenhout et al., 2015; Crippa et al., 2018). Total global emissions of $SO_2$ increased to a peak value of approximately $180\,Tg\,SOx$ (as $SO_2$) $y^{-1}$ in the 1970s, but following

emission reduction policies to improve air quality and reduce acid deposition that were implemented in the 1980s (Hoesly et al., 2018), total global emissions had decreased to approximately $120\,Tg\,SOx\,y^{-1}$ by 2015 (Aas et al., 2019). This trend is captured in global models, but there is substantial temporal variation at the regional scale (Aas et al., 2019). Legislation has driven reductions in $SO_2$ emissions and subsequently $SO_4^{2-}$ aerosol across Europe (Torseth et al., 2012) and North America (Sickles II and Shadwick, 2015; Holland et al., 1998). In these regions reductions in $SO_2$ emissions have had important environmental

and health benefits as well as climate impacts. Turnock et al. (2015) found that between 1970 and 2010 surface $SO_4^{2-}$ aerosol reduced by about 70% in the observations and also in the simulations. For the same period, top of atmosphere (TOA) aerosol radiative forcing over this region increased by $>3\,W\,m^{-2}$ in response to these changes in anthropogenic emissions. Similarly Leibensperger et al. (2012) reported that over the USA aerosol radiative forcing decreased by $1.0\,W\,m^{-2}$ in the period from 1990 to 2010. Emission reduction policies in China have been implemented since 2013, which has reduced anthropogenic $SO_2$

emissions (Aas et al., 2019; Zheng et al., 2018; Hoesly et al., 2018; Liu et al., 2018; Krotkov et al., 2016), and subsequently driven decreases in aerosol optical depth (AOD) (Zhao et al., 2017). However, $SO_2$ emissions from India continue to increase (Aas et al., 2019; Liu et al., 2018; Krotkov et al., 2016).

Good representation of the sulphur cycle in models is essential for constraining uncertainties associated with the impacts of

aerosols on the Earth system and thus understanding the global climate. The global atmospheric loading of $SO_2$ is controlled by the emissions (sources) to the atmosphere and the loss processes, which are oxidation to $SO_4^{2-}$, dry deposition and wet deposition. Global scale $SO_2$ emissions are represented in ESMs using emission inventories such as HTAP (Janssens-Maenhout et al., 2015), OMI-HTAP (Liu et al., 2018), EDGAR (Crippa et al., 2018) and CMIP6 (Feng et al., 2020), the latter being developed for use by models participating in the CMIP6 project (Eyring et al., 2016). Although uncertainty in $SO_2$ emissions

is relatively low (Hoesly et al., 2018), in bottom up inventories such as HTAP and EDGAR there may be uncertainty in the

emission and activity factors, in the conversion from country scale to grid scale, and the input data may be incomplete or subject to rapidly changing economic and/or policy conditions (Janssens-Maenhout et al., 2015). In satellite derived data sets there is uncertainty associated with the retrieval methods and the signal to noise ratio, which can make smaller sources and background concentrations more difficult to detect (Fioletov et al., 2016). Yang et al. (2019) have also found that injection height is a larger source of uncertainty in model representation of $SO_2$ emissions than inventory uncertainty, affecting surface concentrations by $70-130\%$ depending on sector and region, compared with $8-14\%$ from inventory uncertainty. The impact of injection height in UKESM1 was demonstrated by Mulcahy et al. (2020) who found that emitting $SO_2$ higher in to the atmosphere, rather than in to the lowest model level increased the burden from $0.53\,\mathrm{Tg}$ to $0.61\,\mathrm{Tg}$ and the lifetime from $2.08$ to $2.21$ days, although $SO_4^{2-}$ was not significantly affected.

Anthropogenic emissions of $SO_2$ are generally from point sources such as power stations or smelters. Once emitted $SO_2$ has a lifetime of approximately two days, although this can vary from $15\,\mathrm{h}$ to $65\,\mathrm{h}$ in summer and winter, respectively (Lee et al., 2011). The lifetime of $SO_2$ depends on both wet and dry deposition of the molecule and the oxidation rate to $SO_4^{2-}$. The $\approx 2$ day lifetime is such that much of the loss via oxidation and deposition occurs locally. $SO_2$ loss near sources and the impact of environmental conditions on loss processes have been investigated in a number of studies.

$SO_2$ deposition is highly dependant on the surface type, soil pH, solar radiation level, near-surface relative humidity and, in particular, whether the underlying surface is wet or dry, with deposition increasing significantly for a wet surface. Wys et al. (1978) calculated diurnal averaged deposition of emitted $SO_2$ onto an agricultural field of $35\%$ within $300\,\mathrm{km}$ of the emission source, with daytime deposition significantly higher than at night. The same study found that $15\%$ of emitted $SO_2$ was dry deposited onto an arid desert surface within $300\,\mathrm{km}$ of the source. Studies over Europe indicate similar rates of deposition and sensitivity to surface type. For example, using flight observations off the East coast of the UK, Smith and Jeffrey (1975) estimated $50\%$ of the $SO_2$ emitted from UK sources was removed from the atmosphere, or converted to $SO_4^{2-}$ by the time it was observed in air parcels over the North Sea. This amounts to a loss or conversion of $50\%$ emitted $SO_2$ within $\approx$ 200-300 km of the emission source. Smith and Jeffrey (1975) further partitioned this loss into $30$-$35\%$ due to dry deposition and $\approx$ 10-15% oxidation to $SO_4^{2-}$, with wet deposition making only a minor contribution to the total loss. Similar rates of $SO_2$ loss have been observed in a number of other observational studies of dry deposition (e.g., Payrissat and Beilke, 1975; Garland, 1977; Garland and Branson, 1977; Fowler, 1978; Erisman and Baldocchi, 1994). Studies analysing $SO_2$ dispersion around U.S. power stations found that fractional oxidation rates to $SO_4^{2-}$ are sensitive to the amount of solar radiation, with rates ranging from a winter low of $1\times10^{-3}\,\mathrm{h}^{-1}$ to a summer high of $1.5\times10^{-2}\,\mathrm{h}^{-1}$ (Altshuller, 1979; Meagher et al., 1983). Representing the $SO_2$ loss processes is challenging for ESMs because $200-300\,\mathrm{km}$ is represented by $1-2$ grid cells, meaning that deposition and oxidation are parameterized on the model grid scale and may not capture temporal and spatial variation. In addition there is uncertainty associated with the oxidation and deposition processes.

In the atmosphere $SO_2$ can be oxidised in the gas phase by hydroxyl (OH) radicals and in the aqueous phase by reactions in cloud and rain water involving hydrogen peroxide ($H_2O_2$), ozone ($O_3$), $O_2$ catalyzed by transition metal ions, and other oxidants, to form $SO_4^{2-}$ (see Turnock et al. (2019) and references within. The oxidation chemistry is necessarily simplified in many models due to the computational cost of detailed chemistry schemes, but studies have shown that oxidant levels can impact the lifetime of aerosol precursor species and ultimately global radiative forcing (Mulcahy et al., 2020; Karset et al., 2018). Uncertainty in aerosol radiative forcing also results from different values of cloud water pH, which alters $SO_4^{2-}$ formation by changing the rate of aqueous phase oxidation of $SO_2$ by ozone (Turnock et al., 2019). Observations have shown that cloud pH is both temporally and spatially variable (Aleksic et al., 2009; Murray et al., 2013; Schwab et al., 2016; Li et al., 2017), although measurements are very sparse. Typically, this variation is not accounted for in global chemistry climate models, including UKESM1 and its predecessor HadGEM3-GC3.1, both of which use a temporally and spatially constant cloud pH of 5.0. Turnock et al. (2019) found that increasing the cloud pH by 1.0 in HadGEM3-GC3.1 reduced the total $SO_2$ column by up to 50% over Europe, North America, and East Asia for 1970–1974 and 2005–2009. The impact on $SO_4^{2-}$ was variable due to the different $SO_2$ loadings over the different regions and in the different time periods. Overall aerosol radiative forcings varied by up to $4\,W\,m^{-2}$, with larger changes in some regions depending on whether cloud water pH was assumed to have increased or decreased over recent decades.

Loss of $SO_2$ and $SO_4^{2-}$ to the Earth's surface by deposition can be through dry or wet processes. Dry deposition describes the removal of a gas or particle through direct contact of air with the Earth's surface and wet deposition describes the incorporation of gases or particles into rain droplets or snow crystals and their subsequent removal through precipitation. Globally dry deposition removes around 45% of $SO_2$ from the atmosphere (Chin et al., 2000). The importance of dry deposition in the global sulphur budget is the reason why we target it for development in UKESM1. Dry deposition of $SO_2$ in ESMs is generally represented by a resistance in series approach (e.g. Archibald et al., 2020; Wu et al., 2020). Deposition of $SO_4^{2-}$ is mainly via wet processes (approximately 90%, Chin et al., 2000), including nucleation scavenging within the cloud (rain out) and impact scavenging below the cloud (wash out), but dry deposition of $SO_4^{2-}$ does occur through gravitational settling. Deposition processes are necessarily parametrized in global models because they occur at sub-grid scales and this contributes to model uncertainty. Further, observational flux data sets are sparse and frequently temporally and spatially limited, hindering model evaluation of deposition processes at regional to global scales.

Sulphur species are relatively well observed compared to many atmospheric components as their role in air pollution is well established. In the 1970's and 1980's the increasingly detrimental impacts of rising $SO_2$ emissions on acid deposition, air quality and human health in Europe and North America led to monitoring networks being set up in these regions (Torseth et al., 2012; MACTEC-Engineering and Consulting, 2005). Rising pollution in Asia also led to the establishment of the The Acid Deposition Monitoring Network in East Asia (EANET) in 2001 (e.g. Wang et al., 2008). However, even with these data sets it is only possible to evaluate model simulations of the recent historical period and similar data sets are not available for other large source regions such as India, the Middle East, or remote regions. Further, the lack of a range of measurements, including

flux observations, hinders detailed process studies at large scales. Since the early 2000's satellite observations of near surface $SO_2$ have also become available. Of these, the satellite data sets with the best temporal resolution and spatial coverage for $SO_2$ are from the Ozone Monitoring Instrument aboard the NASA Earth Observing System Aura spacecraft (Fioletov et al., 2016). Although biases in the $SO_2$ retrieval from OMI limit its use at high and low latitudes in winter and over areas with low atmospheric $SO_2$ loading, they do provide valuable information over regions where there are no long term, or even any

ground-based observations (Li et al., 2020; Levelt et al., 2018).

This paper is configured as follows; the model (UKESM1), the model simulations, observation data sets and modifications to UKESM1's $SO_2$ dry deposition parameterization are described in Section 2. In Section 3 we evaluate UKESM1 against observations of surface $SO_2$ and $SO_4^{2-}$ and total column $SO_2$. In Section 4 we assess the impact of the modifications to UKESM1's

$SO_2$ dry deposition parameterization. The discussion and conclusions are presented in Sections 5 and 6.

## 2  Methods

### 2.1  UKESM1

UKESM1 is the latest generation Earth System (ES) model developed in the UK. UKESM1 has HadGEM3-GC3.1 (Kuhlbrodt

et al., 2018; Williams et al., 2018) as its physical-dynamical core. HadGEM3-GC3.1 is comprised of the Global Atmosphere 7.1 (GA7.1) configuration of the Unified Model (UM) (Walters et al., 2019; Mulcahy et al., 2018); the Nucleus for European Modelling of the Ocean (NEMO) model (Storkey et al., 2018); the Los Alamos Sea Ice Model (CICE, Ridley et al., 2018) and the Joint UK Land Environment Simulator (JULES) land surface model (Best et al., 2011). The additional ES process models include the stratospheric-tropospheric (StratTrop) version of the United Kingdom Chemistry and Aerosol (UKCA) model

(Archibald et al., 2020), the Model of Ecosystem Dynamics, nutrient Utilisation, Sequestration and Acidification, (MEDUSA, Yool et al., 2013) and the terrestrial biogeochemistry component of JULES (Clark et al., 2011). UKESM1 is described in detail, along with its component models and the coupling between them, by Sellar et al. (2019). The aerosol scheme used in UKESM1 (GLOMAP-Mode, Mann et al., 2010), including $SO_2$ emissions and chemistry, is described in detail by Mulcahy et al. (2020). In UKESM1 the land and atmosphere share a regular latitude–longitude grid with a resolution of $1.25° \times 1.875°$

(approximately $135\,\mathrm{km}$ at the mid latitudes). There are 85 vertical levels on a terrain following hybrid height coordinate with a model lid at $85\,\mathrm{km}$ above sea level and 50 of these levels below $18\,\mathrm{km}$. The ocean has a horizontal resolution of $1°$ and 75 vertical levels. While the atmospheric time step of the model physics is 20 minutes, due to the inherent computational cost of the chemistry and aerosol components, both of these components are called once per hour.

In UKESM1 the $SO_2$ emissions, including anthropogenic sources are from the CMIP6 inventory (Feng et al., 2020). Large explosive volcanic sources and biomass burning sources are not interactively modelled, but prescribed using the CMIP6 stratospheric aerosol climatology (Sellar et al., 2019) and van Marle et al. (2017) emissions inventory respectively. Continuously

degassing volcanic sources are also included as present-day, three-dimensional, temporally fixed (i.e. no seasonal variation) fields (Dentener et al., 2006). Emissions from the energy and industrial sectors are all emitted into the first model layer. We

summarize how loss of $SO_2$ from the atmosphere via oxidation, wet and dry deposition is modelled here, but for a detailed description of these processes in UKESM1 the reader is referred to Archibald et al. (2020) and Mulcahy et al. (2020). Gas- and aqueous-phase oxidation of $SO_2$ to $SO_4^{2-}$ is represented by the reactions shown in Table 1 (Pham et al., 1996; Sander et al., 2003; Kreidenweis et al., 2003). Dry deposition of $SO_2$ is parameterized following the resistance in series approach originally developed by Wesely (1989) (see Section 2.2.1). Loss via wet deposition is the $SO_2$ that is scavenged and subse-

quently converted to $SO_4^{2-}$ in rainwater. It is parameterized as a first-order loss rate, calculated as a function of UKESM1's three-dimensional convective and large-scale precipitation (Archibald et al., 2020; O'Connor et al., 2014). Sulphate aerosol is also removed from the atmosphere by dry and wet deposition (Mulcahy et al., 2020). The aerosol dry deposition and sedimentation are represented by a resistance in series approach similar to that used for gaseous species, but which also accounts for aerosol size (Mann et al., 2010). Wet deposition is parameterized in UKESM1 by an in-cloud convective plume scavenging

scheme following the approach described by Kipling et al. (2013) and by nucleation scavenging (Mulcahy et al., 2020).

**Table 1.** Summary of $SO_2$ oxidation chemistry in UKESM1

| Gas phase reactions |
| --- |
| $SO_2 + OH \rightarrow SO_3 + HO_2$ |
| $SO_2 + O_3 \rightarrow SO_3$ |
| $SO_3 + H_2O \rightarrow H_2SO_4 + H_2O$ |
| **Aqueous phase reactions** |
| $HSO_3^- + H_2O_2 \rightarrow SO_4^{2-}$ |
| $HSO_3^- + O_3 \rightarrow SO_4^{2-}$ |
| $SO_3^{2-} + O_3 \rightarrow SO_4^{2-}$ |

## 2.2   $SO_2$ dry deposition parameterization

The UKESM1 parameterization of $SO_2$ dry deposition follows that described in Wesely (1989). This scheme uses the widely accepted approach of calculating the flux of a depositing gas as a function of a deposition velocity multiplied by the concen-

tration gradient of the gas between a reference height ($z$, e.g. the lowest model level) and the receptor surface (Eqn. 1. The deposition velocity is calculated by analogy with electrical resistance and is inversely proportional to three resistances to deposition, representing the three stages of gaseous transport to a receptor surface. These are: (i) aerodynamic resistance ($R_a$) to gas transport through the near-surface turbulent layer, (ii) viscous resistance to gas transfer across a quasi-laminar layer surrounding

the receptor surface ($R_b$) and (iii) structural resistance to deposition of the receptor surface itself ($R_c$). For a detailed description of this approach, see Wesely (1989); Erisman and Baldocchi (1994); Zhang et al. (2003). The deposition velocity is calculated for each fractional surface type in a given model grid box, as is a resulting loss rate (flux) of $SO_2$ from the atmosphere. The loss rates to each fractional surface are combined, resulting in the total loss of $SO_2$ from the model atmosphere due to dry deposition. The deposition velocity is given by Equation 2. If the surface is covered by vegetation, $R_a$ is generally calculated at a zero-plane displacement height $z = z - d$, where d is usually 0.6-0.8 times the vegetation height in metres. The UKESM1 calculation of $R_a$ and $R_b$ follow standard approaches (see Eqn. 3 and 4). The aerodynamic resistance, $R_a$, is calculated from the wind profile taking into account atmospheric stability and the surface roughness, where $z0$ is the roughness length, $\psi$ is the Businger dimensionless stability function, $\kappa$ is von Karman's constant, and $u_*$ is the friction velocity. The quasi-laminar sub-layer resistance, $R_b$, is calculated with $Sc$ the Schmidt number, and $Pr$ the Prandtl number.

The surface or canopy resistance to deposition, $R_c$, is the most difficult of the three resistances to parameterize as it is sensitive to biochemical details of the individual receptor surfaces. $R_c$ is typically a function of the following receptor-specific resistances: (i) canopy stomatal resistance ($R_{stom}$) combined with the mesophyll resistance ($R_m$) of a given plant, (ii) canopy cuticle or external leaf resistance ($R_{cut}$) and (iii) soil resistance ($R_{soil}$), combined with an in-canopy resistance ($R_{inc}$), describing the turbulent transport of a gas through the plant foliage to the ground. The stomatal resistance, leaf cuticle resistance and soil resistance are assumed to operate in parallel. For surfaces not covered with vegetation (e.g. open water, bare soil or snow covered surfaces), $R_c$ is made equal to one of; $R_{water}$, $R_{soil}$, $R_{snow}$. The receptor-specific resistances are combined as shown in Equation 5 to calculate $R_c$. In UKESM1, $R_{stom}$ follows the approach outlined in Wesely (1989), based on the original work of Baldocchi et al. (1987). $R_{stom}$ is first calculated for water vapour for each vegetation type, $R_{stom}$ for other gases is then derived by scaling $R_{stom}$ for water vapour by the ratio of the diffusion coefficient for the gas in question and that of water vapour. Due to a general lack of knowledge $R_m$ values are assumed to be zero for all gases. In UKESM1 $R_{inc}$ and $R_{soil}$ are combined into a single value (referred to hereafter as $R_{soil}$). In UKESM1, $R_c$ for $SO_2$ for the 13 fractional land cover types is initially set to the standard surface resistance ($R_{surf}$) values given in Table A2.

$$F = V_d \times C \tag{1}$$

$$V_d = \frac{1}{R_a + R_b + R_c} \tag{2}$$

$$R_a = (ln(\frac{z}{z0}) - \psi)/(\kappa u_*) \tag{3}$$

$$R_b = (Sc/Pr)^{\frac{2}{3}}/(\kappa u_*) \tag{4}$$

$$R_c = [\frac{1}{R_{stom} + R_m} + \frac{1}{R_{inc} + R_{soil}} + \frac{1}{R_{cut}}]^{-1} \tag{5}$$

### 2.2.1 Modifications to UKESM1's $SO_2$ dry deposition parameterization

In this study we investigate two changes to the $SO_2$ dry deposition parameterization in UKESM1. Firstly, we account for a key omission in UKESM1 in that for $R_{cut}$ and $R_{soil}$ no account is taken as to whether the receptor surface is wet or dry, nor of the near surface relative humidity. Observational studies suggest that $SO_2$ dry deposition (through a decrease in $R_c$) is significantly more efficient over wet surfaces compared to dry surfaces, as well as for increasing values of near surface relative humidity due to the high solubility of $SO_2$ in water (e.g., Garland and Branson, 1977; Fowler, 1978; Erisman and Baldocchi, 1994; Erisman et al., 1994). We apply the findings from these studies to extend the calculation of $R_{cut}$ for $SO_2$ in UKESM1 to be a function both of whether the model vegetation is wet or dry and to the near surface relative humidity. This change allows a surface to remain wet after rainfall for a period of three hours, where previously it would have been "dry" immediately after the rainfall event. $R_{soil}$ for $SO_2$ is also made a function of near surface relative humidity. These changes are referred to as **$R_{surf}$-mod** and will impact $SO_2$ dry deposition over land surfaces. We include a more detailed description of the modifications to UKESM1's $SO_2$ parameterization in Appendix A. Secondly, we change the the surface resistance term for $SO_2$ dry deposition to water ($R_{water}$) from an erroneously high value of $148 \, s \, m^{-1}$ to $1 \, s \, m^{-1}$ to better reflect the high solubility of $SO_2$ in water. While lower than the value of $20 \, s \, m^{-1}$ used by Zhang et al. (2003), it reflects the small, observed value of $0.004 \, s \, m^{-1}$ from Garland (1977). This change is referred to as **$R_{water}$-mod** and will impact $SO_2$ dry deposition predominantly over the ocean.

In addition to the primary changes to the $SO_2$ dry deposition parameterization (**$R_{water}$-mod** and **$R_{surf}$-mod**), we also include two secondary modifications. These are (1) an update in the calculation of the stability parameter (*z/L*) to better describe dry deposition in very stable atmospheric conditions, and (2) a bug fix in the dimethyl sulphide (DMS) chemistry. The stability parameter (*z/L*) describes the flux profile relationship and is important for calculating $R_a$ in Equation 3. Note that the Monin-Obhukov length (*L*) is derived locally in the UKCA code using local values of air density, temperature and friction velocity, where the friction velocity is computed in the UM turbulence scheme and so is consistent across subroutines. Here we update the calculation of the stability parameter from that given by Dyer (1974) to that described by Holtslag and Bruin (1988). We also reduce the reference height for dry deposition (*z*) from $50 \, m$ to $10 \, m$. The reference height is the height below which there is no turbulence in very stable conditions and is also important for calculating $R_a$. Following Ganzeveld and Lelieveld (1995) the reference height should be half the average height of the lowest model layer, which in UKESM1 is $20 \, m$. The changes to *z/L* and *z* act to reduce the rate at which the deposition velocity decreases in very stable conditions, although we note that there is also an impact on the calculation of aerodynamic resistance in unstable conditions. The DMSO bug fix corrects the equation for dimethyl sulphide (DMS) oxidation by OH (see Reaction R1) in UKCA's StratTrop mechanism, where the products incorrectly contain more sulphur atoms than the reactants. We substitute Reaction R1 with Reactions R2 and R3. This reduces the $SO_2$ yield to a maximum of 0.84, which may be further reduced as DMSO deposits to the Earth's surface. However, the changes in simulated $SO_2$ are actually only of the order of 1% because anthropogenic sources are not affected by this change.

Although the secondary changes incorporate important updates in to the model, their impact on the atmospheric $SO_2$ loading in UKESM1 is small in comparison to that driven by **$R_{water}$-mod** and **$R_{surf}$-mod** and we do not discuss it here.

$$C_2H_6S + OH \rightarrow SO_2 + CH_3SO_2OH \tag{R1}$$

$$C_2H_6S + OH \rightarrow 0.6SO_2 + 0.4C_2H_6SO_2 + CH_3O_2 \tag{R2}$$

$$C_2H_6SO_2 + OH \rightarrow 0.6SO_2 + 0.4CH_3SO_2OH. \tag{R3}$$

## 2.3 Model simulations

For this evaluation we initially use 4 simulations from the 19 member ensemble of historical simulations that were conducted
for UKESM1's contribution to CMIP6 (Sellar et al., 2019; Tang, 2019). The historical simulations cover the period from 1850 to the end of 2014, thus modelling the evolution of climate and composition since the pre-industrial era. These simulations are forced by transient external forcings of solar variability, land use, well-mixed greenhouse gases and other trace gas emissions and aerosols. The volcanic forcing due to the stratospheric injection of $SO_2$ from volcanic eruptions is prescribed as a zonal mean climatology of the stratospheric aerosol optical properties over the historical period. All forcings and how they are
implemented in UKESM1 are described fully in Sellar et al. (2019). Each historical ensemble member was initialised from a different date in the pre-industrial control simulation (Yool et al., 2020). We use monthly mean output for surface $SO_2$ and $SO_4^{2-}$ concentrations, and $SO_2$ dry deposition flux. We use a second four member ensemble of historical simulations to evaluate the impact of the changes to the $SO_2$ dry deposition parameterization described in Section 2.2.1, hereafter this ensemble is referred to as UKESM1-SO2. The UKESM1-SO2 historical simulations are set up and run as for the UKESM1 historical
simulations. To calculate the detailed $SO_2$ budget we utilise the atmosphere-only configuration of UKESM1 and UKESM1-SO2. This is referred to as the Atmospheric Model Intercomparison Project (AMIP) configuration and allows us to generate the diagnostics required for the budget analysis that were not output in the historical simulations at a reduced computational cost. The UKESM1 AMIP configuration is driven by observed sea surface temperature (SST) and sea ice. It does not include the additional dynamic ocean and land surface components (Eyring et al., 2016). Instead, the required vegetation (vegetation
fractions, leaf area index, canopy height) and surface ocean biology fields (DMS and chlorophyll) are taken from a single UKESM1 historical member and are prescribed as ancillary data, thereby maintaining traceability to the fully coupled model. For the $SO_2$ budget calculations AMIP simulations were run from 1979 to the end of 1983.

**Table 2.** Summary of model configurations used in this study

|  | UKESM1 | UKESM1-SO2 |
|---|---|---|
| Configuration | Historical | Historical |
| No. of members | 4 | 4 |
| Modifications | - | (1) $R_{water}$-mod and $R_{surf}$-mod |
|  |  | (2) Update to $z/L$ and $z = 10\,\text{m}$ |
|  |  | (3) DMSO chemistry bug fix |
| Configuration | AMIP | AMIP |
| No. of members | 1 | 1 |
| Modifications | - | (1) $R_{water}$-mod and $R_{surf}$-mod |
|  |  | (2) Update to $z/L$ and $z = 10\,\text{m}$ |
|  |  | (3) DMSO chemistry bug fix |

## 2.4 Ground based observations

We compare the modelled surface $SO_2$ and $SO_4^{2-}$ concentrations to observations from the Clean Air Status and Trends Network (CASTNet, http://epa.gov/castnet/javaweb/index.html, Finkelstein et al., 2000) and the European Monitoring and Evaluation Program (EMEP, http://ebas.nilu.no/; Torseth et al., 2012). CASTNet provides surface observations of mean seasonal $SO_2$ and $SO_4^{2-}$ concentrations which are available from 1987 to the present at 97 sites situated in the United States of America (USA). In this study we used observations from the CASTNet sites designated as "western reference" or "eastern reference".

The reference sites have been reporting measurements since at least 1990 and are used for determining long term trends, (e.g., Clarke et al., 1997; Holland et al., 1998; MACTEC-Engineering and Consulting, 2005; Baumgardner et al., 2002). There are 16 western reference sites and 33 eastern reference sites which are located in the continental USA to the west and east of $100°$W respectively (MACTEC-Engineering and Consulting, 2005). The eastern region is significantly more polluted than the western region due to the larger number of $SO_2$ sources there. We therefore keep the western and eastern data sets separate

to assess how UKESM1 performs in the two regions. Hereafter, we refer to the eastern and western USA regions as USA–E and USA–W, respectively. For this evaluation we used the mean seasonal surface concentrations for $SO_2$ and $SO_4^{2-}$ which are measured with filter pack samplers at weekly sampling intervals. Details of the quality control procedures and of how the mean seasonal concentrations are calculated are given in (Baumgardner et al., 2002).

We also evaluate simulated $SO_2$ dry deposition flux from UKESM1 against observations from CASTNet, using the same eastern and western reference sites that were used to evaluate surface $SO_2$ and $SO_4^{2-}$ concentrations. The CASTNet deposition fluxes are derived using modelled deposition velocities rather than directly measured fluxes, which are difficult to obtain due to the requirement for extensive instrumentation and technical resource. Direct measurements of $SO_2$ dry deposition flux are

therefore temporally and spatially limited, and not suitable for evaluating long term trends. To derive the $SO_2$ dry deposition fluxes, measurements of $SO_2$ concentration are combined with routine meteorological measurements, information on the land use type and LAI at the measurement site. This data is then combined with modelled deposition velocities from the Multi Layer Model (MLM, Meyers et al., 1998; Saylor et al., 2014). The methodology used to derive the $SO_2$ dry deposition fluxes for CASTNet is described in Clarke et al. (1997) and Baumgardner et al. (2002). While the modelled $SO_2$ dry deposition fluxes can be under-predicted by approximately 30% (Clarke et al., 1997), it is considered to be the best available approach to regional scale assessment of dry deposition (Finkelstein et al., 2000; Baumgardner et al., 2002; E. Sickles and Shadwick, 2007; Sickles and Shadwick, 2007). This approach has been used to determine $SO_2$ dry deposition fluxes for CASTNet since 1987 (e.g., Clarke et al., 1997; Baumgardner et al., 2002) and to assess global and regional scale models (Vet et al., 2014; Tan et al., 2018; Tang et al., 2018).

Surface $SO_2$ and $SO_4^{2-}$ concentrations have been monitored at EMEP sites for the period $1972 - $ present (Torseth et al., 2012). In this study we have used observations of surface concentration of $SO_2$ and $SO_4^{2-}$ from 48 and 42 sites respectively. We have selected sites where there are at least 10 years of continuous measurements and with a few exceptions have used sites where $SO_2$ and $SO_4^{2-}$ were co-located. We use monthly mean observations for both species. No $SO_2$ dry deposition data were available from EMEP. The locations of the CASTNet and EMEP sites used in this study are shown in Figure 1.

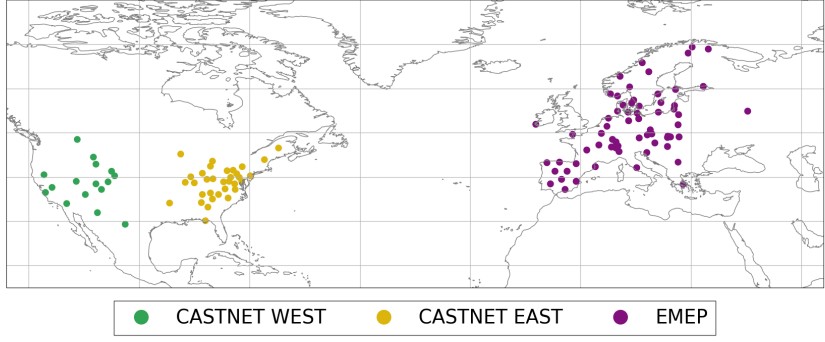

**Figure 1.** Map of the locations of the CASTNet and EMEP measurement sites used in this study.

## 2.5  Data processing

For this evaluation we calculated seasonal averages for the modelled surface $SO_2$ and $SO_4^{2-}$ concentrations and for the EMEP observational data. The seasonal periods were defined as December-January-February (DJF), March-April-May (MAM), June-July-August (JJA) and September-October-November (SON). The CASTNet data for all variables was available as seasonal averages for these periods. Model grid cell output was co-located with the CASTNet and EMEP measurement sites. In some

cases, this resulted in model data from a particular grid cell being compared with more than one measurement site. For the time series analysis, regional means and standard deviations were calculated across the sites in the USA-E, USA-W and Europe regions. Although there is spatial variation in the surface $SO_2$ and $SO_4^{2-}$ concentrations across Europe, for example concentrations are relatively low in Scandinavia, but are much higher in South East Europe, it is less easy to classify "clean" and "polluted" regions at the global model scale. Therefore we classify Europe as a single region. For the spatial analysis and calculation of time series statistics, we calculate mean values over the whole time series, i.e. 1987-2014 and for two time slices at the start (1990-1995) and end of the time series (2009-2014). For the time slices we only used sites that had at least three out of five years data available. We investigate the two time slices to assess the model's performance during the different pollution levels at the start and end of the time series. We determine the rate of change (trend) in the surface concentrations by calculating the linear regression for 1987-2014, 1990-1995 and 2009-2014.

## 2.6 Satellite observations

Total column $SO_2$ (TCSO$_2$) measurements came from the Ozone Monitoring Instrument (OMI) and were obtained from the Goddard Earth Sciences Data and Information Services Centre (https://aura.gesdisc.eosdis.nasa.gov/data/Aura_OMI_Level2/ OMSO2.003) (Li et al., 2020). OMI is situated on-board NASA's polar-orbiting Aura satellite launched in 2004 with a local overpass time of approximately 13:45. OMI has a nadir footprint of $13\,km \times 24\,km$ and a spectral viewing range of 270 to 500 nm (Levelt et al., 2018). The TCSO$_2$ product is quality controlled for cloud radiation fraction $> 0.0$ and $< 0.5$, solar zenith angle $< 65°$, the South Atlantic Anomaly flag $= 0$, ice cover flag $= 0$, the air mass factor (AMF) $> 0.3$ and TCSO$_2$ $> -1.0$ Dobson unit (DU). Background TCSO$_2$ average values tend to be positive near-zero quantities (i.e. just above 0.0), where some soundings are slightly negative. If only positive TCSO$_2$ were incorporated in the background averages, this would positively skew the true value.

For a robust comparison between model simulations and satellite data both data sets typically require spatio-temporal co-location to reduce sampling (representation) errors. To achieve this, high temporal resolution (e.g. 3 hourly or 6 hourly) model output of 3D tracer and pressure fields are required over the analysis period to capture e.g. diurnal variability (Pope et al., 2016; Monks et al., 2017). However, this is difficult when using standard climate model simulations, including those used in this study, which typically output monthly means due to their long term climate focus. In this comparison we performed tests to show that using the monthly mean model output was suitable given the relatively uniform diurnal cycle of $SO_2$ emissions. For these tests we made initial comparisons of model output and satellite data using 6 hourly and monthly mean output from the same UKESM1 simulation. We found that the temporal sampling of the model was not overly critical for $SO_2$, i.e. modelled $SO_2$ has a sufficiently long lifetime to dampen the influence of diurnal sampling of the model. Further details on the TCSO$_2$ product, how it was processed to obtain TCSO$_2$ values and the assessment of the temporal resolution is given in Pope and Chipperfield (2021).

# 3 Evaluation of trends and biases in modelled $SO_2$ and $SO_4^{2-}$ concentrations

## 3.1 Time series analysis of surface concentrations of $SO_2$ and $SO_4^{2-}$

UKESM1 simulations of surface of $SO_2$ and $SO_4^{2-}$ concentrations are compared with observations from the CASTNet and EMEP networks for the period 1987–2014 in Figure 2. The statistics summarizing the model bias and trends over this period as well as for the 1990–1995 and 2009–2014 time slices are shown in Tables 3 and 4. We find that UKESM1 captures the historical reduction in surface $SO_2$ and $SO_4^{2-}$ concentrations. This is in agreement with Aas et al. (2019) who reported that an ensemble of global aerosol models were generally able to capture the recent historical declines in these two species over the USA and Europe. UKESM1 over-predicts surface $SO_2$ concentrations in all three regions, but the direction of the model's bias in surface $SO_4^{2-}$ concentrations is spatially variable.

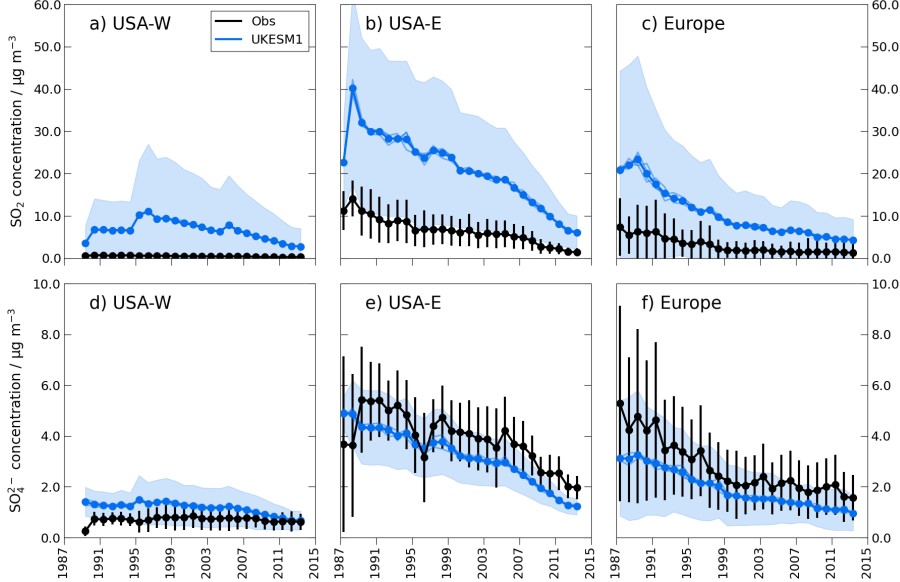

**Figure 2.** Time series of observed and modelled mean annual surface $SO_2$ (top row) and $SO_4^{2-}$ (bottom row) concentrations for USA–W (a, d; $N$ sites = 16), USA–E (b, e; $N$ sites = 33) and Europe (c, f; $N$ sites = 48 for $SO_2$ and 42 for $SO_4^{2-}$). Each point in the time series represents the mean across the measurement sites in the region. Note that the vertical scale for $SO_2$ (a-c) is a factor of 6 larger than that for $SO_4^{2-}$ (d-f).

Figures 2a-c show that in both Europe and USA–E the model over-predicts surface $SO_2$ concentrations, particularly at the start of the time series, which then decrease too rapidly. Over Europe, the observed surface $SO_2$ concentrations decrease at a rate of $0.72\,\mu g\,m^{-3}\,y^{-1}$ in the period 1990–1995 which slows to $0.38\,\mu g\,m^{-3}\,y^{-1}$ by 2009–2014. However, the modelled surface $SO_2$ concentrations decrease by $2.52\,\mu g\,m^{-3}\,y^{-1}$ for 1990–1995, slowing to only $1.31\,\mu g\,m^{-3}\,y^{-1}$ by 2009–2014. Over USA–E the observed surface $SO_2$ concentrations decrease at similar rates at the start and end of the time series. UKESM1 is

better able to capture the trend at the start of the time series, but simulates a too rapid reduction in surface $SO_2$ concentrations after 2005 (see Figure 2b). Over USA–E UKESM1 simulates the sharp drop in surface $SO_2$ concentrations that occurred in 1995 following the implementation of Phase 1 of the USA's Clean Air Act Amendments (McHale et al., 2021). However, the model then simulates relatively high surface $SO_2$ concentrations for the period 1996–1999, rather than the sustained lower surface $SO_2$ concentrations that are observed after 1995. Over USA–W the observed surface $SO_2$ concentrations remain steady

from 1987 to 2014 due to there being fewer sources in this region. Figure 2a shows that UKESM1 simulates the steady surface $SO_2$ concentrations at the start of the time series, albeit with a positive bias. However, after 1995 UKESM1 simulates decreasing surface $SO_2$ concentrations in USA–W, which brings the modelled values in to better agreement with the observations, but introduces an artificial trend into the modelled time series.

UKESM1 over-predicts the annual mean surface $SO_2$ concentrations in the polluted regions of Europe and USA–E by a factor of 3.2 to 3.4 over the period 1987–2014, although the absolute bias is higher USA–E (see Table 3). While the absolute magnitude of the bias in mean annual surface $SO_2$ concentration is less in USA–W compared with the polluted regions, proportionally it is much larger, with the model simulating surface $SO_2$ concentrations more than 10 times the observed values. The absolute magnitude of the bias in mean annual surface $SO_2$ concentration decreased from 1990–1995 to 2009–2014 in all

three regions (see Table 3) reflecting the model's more rapid decrease in surface $SO_2$ concentrations relative the observations. However, the NMB values were slightly higher in 2009–2014 compared with 1990–1995.

Figures 2e-f show that UKESM1 captures both the magnitude and trends in surface $SO_4^{2-}$ concentrations better than the surface $SO_2$ concentrations. The model simulated surface $SO_4^{2-}$ concentrations decreasing at a rate of $0.13\,\mu g\,m^{-3}\,y^{-1}$ (USA–E)

and $0.09\,\mu g\,m^{-3}\,y^{-1}$ (Europe) compared with the observed trend of $\approx 0.10\,\mu g\,m^{-3}\,y^{-1}$ (see Table 4). UKESM1 does under-predict mean annual surface $SO_4^{2-}$ concentrations in the polluted regions of USA–E and Europe, but the model bias is relatively small compared with the large over prediction of mean annual surface $SO_2$ concentration (see Tables 3 and 4). We also find that there is a large range associated with the modelled and observed data, and that the mean surface $SO_4^{2-}$ concentrations lie within these ranges. The model bias remained relatively constant over the period from 1987–2014, ranging from $-0.96\,\mu g\,m^{-3}$

to $-0.80\,\mu g\,m^{-3}$ for the periods 1990–1995 and 2009–2014 over USA–E and from $-0.91\,\mu g\,m^{-3}$ to $-0.69\,\mu g\,m^{-3}$ for the same periods over Europe (see Table 4). The picture is different in USA–W where, in contrast to USA–E and Europe, UKESM1 over predicts mean annual surface $SO_4^{2-}$ concentration by an average of 150% for 1987–2014. Both the absolute model bias and the NMB is worse in 1990–1995 than in 2009–2014, which may be attributed to the model simulating a much faster decrease in mean annual surface $SO_4^{2-}$ concentrations compared to the observed trend for the later period (see Table 4).


## 3.2   Spatial evaluation of surface $SO_2$ and $SO_4^{2-}$ concentrations

Figures 3 and 4 show the spatial distribution of modelled and observed mean annual surface $SO_2$ and $SO_4^{2-}$ concentrations for the periods 1990–1995 and 2009–2014 for the USA and Europe. We find that UKESM1 captures the spatial distribution

**Table 3.** Statistics for mean annual surface $SO_2$ concentrations at USA–W, USA–E and Europe. The mean and trend values are in $\mu g\,m^{-3}$ and $\mu g\,m^{-3}\,y^{-1}$ respectively.

| | | 1987–2014 | 1990–1995 | 2009–2014 |
|---|---|---|---|---|
| **USA–W** | Mean (obs) | 0.48 | 0.67 | 0.29 |
| | Mean (model) | 6.48 | 6.67 | 3.19 |
| | Bias | 6.00 | 6.00 | 2.90 |
| | NMB | 12.54 | 9.00 | 10.04 |
| | R | 0.20 | 0.89 | 0.93 |
| | Trend (obs) | $-7.13\times10^{-3}$ | $-1.46\times10^{-2}$ | $-2.05\times10^{-3}$ |
| | Trend (model) | $-4.58\times10^{-2}$ | $-5.74\times10^{-2}$ | -0.36 |
| | N sites | 16 | 6 | 16 |
| **USA–E** | Mean (obs) | 6.34 | 9.09 | 1.87 |
| | Mean (model) | 20.05 | 28.89 | 7.28 |
| | Bias | 14.20 | 19.90 | 5.41 |
| | NMB | 2.41 | 2.20 | 2.92 |
| | R | 0.97 | 0.89 | 0.94 |
| | Trend (obs) | -0.37 | -0.36 | -0.25 |
| | Trend (model) | -1.00 | -0.53 | -1.04 |
| | N sites | 33 | 33 | 33 |
| **Europe** | Mean (obs) | 2.94 | 4.96 | 1.27 |
| | Mean (model) | 10.20 | 15.80 | 4.38 |
| | Bias | 7.26 | 10.90 | 3.12 |
| | NMB | 2.61 | 2.27 | 2.47 |
| | R | 0.93 | 0.99 | 0.97 |
| | Trend (obs) | -0.20 | -0.42 | -0.02 |
| | Trend (model) | -0.66 | -1.35 | -0.27 |
| | N sites | 48 | 43 | 47 |

of surface $SO_2$ and $SO_4^{2-}$ concentrations over each region, simulating higher concentrations in USA–E, central Europe and
eastern Europe where there are numerous large sources, and lower concentrations in USA–W and northern Europe which have much fewer sources (see Figure B1). These figures also show the localised versus dispersed nature of the surface $SO_2$ and $SO_4^{2-}$ concentrations, with high $SO_2$ concentrations located within $2-3$ grid boxes ($200-400\,km$) of the emission sources (see Figure B1), while $SO_4^{2-}$ is distributed more widely. Figure 3 shows that modelled and observed surface $SO_2$ and $SO_4^{2-}$ concentrations across the USA are lower in 2009–2014 compared with 1990–1995 demonstrating the widespread impact of

**Table 4.** Statistics for mean annual surface $SO_4^{2-}$ concentrations at USA–W, USA–E and Europe. The mean and trend values are in $\mu g\,m^{-3}$ and $\mu g\,m^{-3}\,y^{-1}$ respectively.

| | | 1987–2014 | 1990–1995 | 2009–2014 |
|---|---|---|---|---|
| **USA–W** | Mean (obs) | 0.74 | 0.73 | 0.62 |
| | Mean (model) | 1.15 | 1.27 | 0.73 |
| | Bias | 0.44 | 0.54 | 0.11 |
| | NMB | 0.72 | 0.73 | 0.18 |
| | R | 0.14 | 0.7 | 0.90 |
| | Trend (obs) | $-1.01\times10^{-2}$ | $-3.41\times10^{-3}$ | $-9.45\times10^{-3}$ |
| | Trend (model) | $-5.83\times10^{-3}$ | $-1.41\times10^{-2}$ | $-3.97\times10^{-2}$ |
| | N sites | 16 | 6 | 16 |
| **USA–E** | Mean (obs) | 3.82 | 5.17 | 2.18 |
| | Mean (model) | 3.14 | 4.21 | 1.38 |
| | Bias | -0.67 | -0.96 | -0.80 |
| | NMB | -0.19 | -0.19 | -0.37 |
| | R | 0.98 | 0.87 | 0.94 |
| | Trend (obs) | -0.10 | -0.13 | -0.2 |
| | Trend (model) | -0.13 | -0.07 | -0.14 |
| | N sites | 33 | 33 | 33 |
| **Europe** | Mean (obs) | 2.76 | 3.81 | 1.75 |
| | Mean (model) | 1.91 | 2.86 | 1.06 |
| | Bias | -0.85 | -0.94 | -0.69 |
| | NMB | -0.31 | -0.24 | -0.39 |
| | R | 0.97 | 0.91 | 0.74 |
| | Trend (obs) | -0.12 | -0.29 | 0.09 |
| | Trend (model) | -0.09 | -0.09 | 0.03 |
| | N sites | 42 | 41 | 34 |

emission reductions policies. However, the disparity between higher concentrations in USA–E and lower concentrations in USA–W is still apparent for both species in the later period.

In Europe the highest mean annual surface $SO_2$ and $SO_4^{2-}$ concentrations were observed in central and eastern Europe and the south east (SE) of England (see Figure 4). Lower concentrations of both species were observed in northern and western regions, e.g. Scandinavia and the west coast of Ireland. Figure 4c shows that mean annual surface $SO_2$ concentrations were

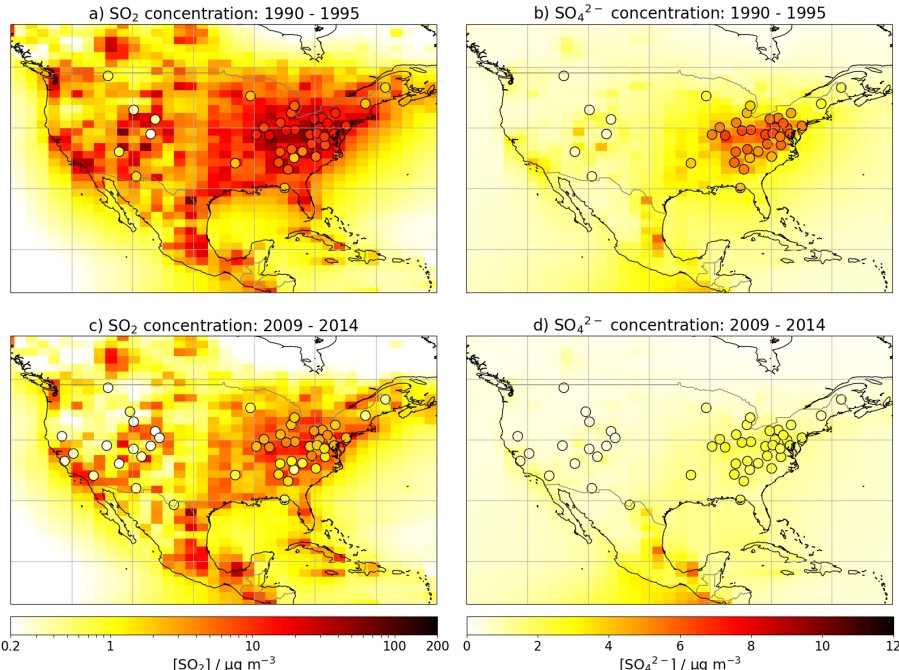

**Figure 3.** Mean annual surface $SO_2$ concentration (a, c) and surface $SO_4^{2-}$ concentration (b, d) for 1990–1995 (top row) and 2009–2014 (bottom row) for modelled output. Observations from 49 CASTNet measurement sites are plotted as black-edged circles on the same colour scale.

generally lower in 2009–2014 compared with 1990–1995, especially in Central and Eastern Europe, due to the impact of air quality legislation. However, for 2009–2014 modelled and observed levels of $SO_2$ remain high in the south eastern region (see Figure 4c). UKESM1 reproduces the spatial distribution of mean annual $SO_2$ concentrations across Europe, but has large positive biases over most of the region. The largest model biases were in eastern and south eastern Europe during the period 1990–1995 where UKESM1 simulates mean annual surface $SO_2$ concentrations of up to $100\,\mu\mathrm{g\,m}^{-3}$ compared with observed values of $10$ - $30\,\mu\mathrm{g\,m}^{-3}$. Figure 4b shows that UKESM1 captures the spatial distribution between low mean annual surface $SO_4^{2-}$ concentrations in northern and western regions of Europe and high mean annual surface $SO_4^{2-}$ concentrations in central Europe. However, the model under-predicts mean annual surface $SO_4^{2-}$ concentrations at many locations across Europe with the largest biases occurring across Denmark and the regions surrounding the Baltic Sea.

### 3.3 Spatial distribution of model bias in surface $SO_2$ and $SO_4^{2-}$ concentrations

Figures 5 and 6 show that the direction of the model biases, whether poisitive or negative, is generally consistent across a region. UKESM1 over-predicts mean annual surface $SO_2$ concentration for all sites in both the USA–E and USA–W regions, and with the exception of some Scandinavian sites, across Europe. Figures 5 and 6 also show that mean annual surface $SO_4^{2-}$

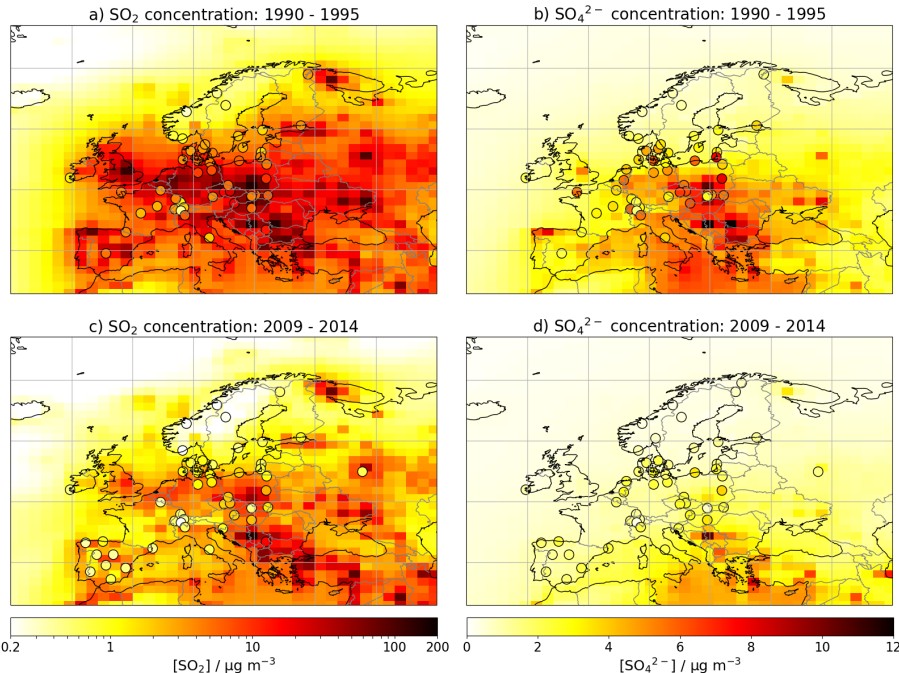

**Figure 4.** Mean annual surface $SO_2$ concentration (a, c) and surface $SO_4^{2-}$ concentration (b, d) for 1990–1995 (top row) and 2009–2014 (bottom row) for modelled output. Observations from EMEP measurement sites ($N$ sites = 48 for $SO_2$ and 42 for $SO_4^{2-}$) are plotted as black-edged circle on the same colour scale.

concentrations are generally under-predicted across the USA–E and European sites, while being over-predicted at the USA–W sites.

The model's over-prediction of mean annual surface $SO_2$ concentration is largest close to the sources. For example, UKESM1 over-predicts surface $SO_2$ concentrations by up to $50\,\mu\mathrm{g\,m}^{-3}$ in the central USA–E area, but only by around $10\,\mu\mathrm{g\,m}^{-3}$ at the surrounding sites. Whilst in Europe the largest biases of around $10\text{-}20\,\mu\mathrm{g\,m}^{-3}$ tend to be in central and eastern areas. The model's tendency to over-predict $SO_2$ concentrations to a greater extent close to the sources is also shown in the plots of NMB (see Figures 5c and 6c) and is likely why there is a larger range in the modelled mean annual surface $SO_2$ concentrations averaged across the USA–E and European measurement sites compared with the observational values (see Figures 2b and c). Figures 5b and 6b show that UKESM1 generally under-predicts surface $SO_4^{2-}$ concentration across the USA–E and European sites, with model biases of -1 to -3 $\mu\mathrm{g\,m}^{-3}$. We find that in USA–E the largest negative biases in surface $SO_4^{2-}$ concentrations are not necessarily co-located with the largest positive biases in $SO_2$, instead occurring at sites several hundred kilometres from the large point sources. Similarly, in Europe the largest biases in surface $SO_4^{2-}$ concentrations occur further north than the large biases in surface $SO_2$ concentration, and in certain sites located near large sources, UKESM1 over-predicts surface $SO_4^{2-}$ concentrations. The plots of NMB show that there is less spatial variation in the model bias for surface $SO_4^{2-}$ con-

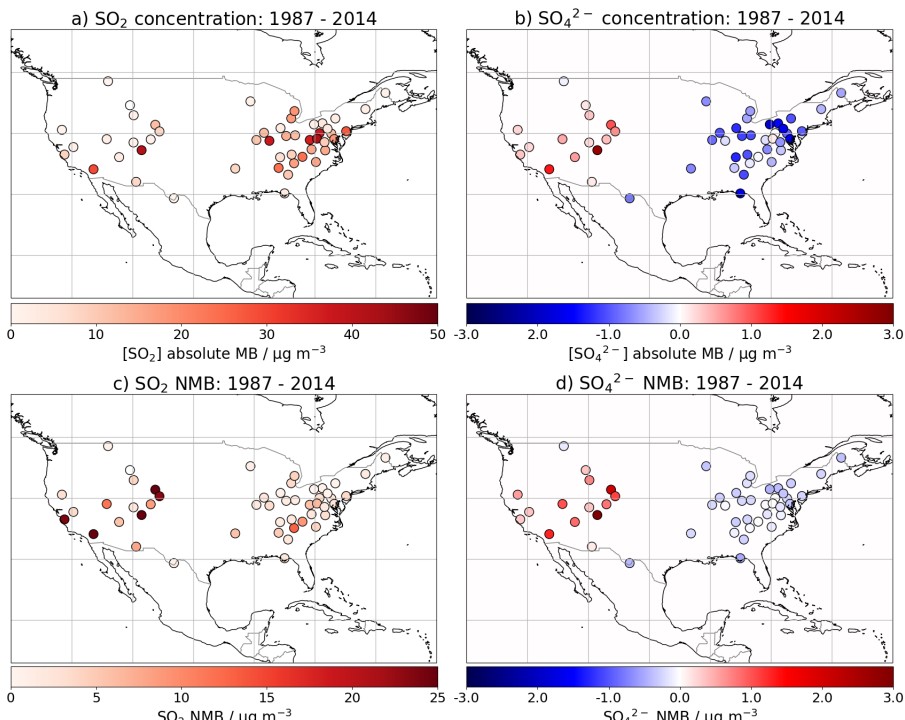

**Figure 5.** Geographic distribution of mean bias (UKESM1 - obs) in mean annual surface $SO_2$ and $SO_4^{2-}$ concentration at 49 CASTNet measurement sites. The mean annual surface concentrations are calculated over the period 1987 to 2014. Absolute mean bias (MB) is shown in (a) and (b) and normalised mean bias (NMB) is shown in (c) and (d). Note that different scales are used for the $SO_2$ model bias (a) and normalised mean bias (c).

centrations reflecting UKESM1's ability to capture the more distributed nature of atmospheric $SO_4^{2-}$. In USA–W UKESM1 over-predicts both $SO_2$ and $SO_4^{2-}$ concentrations at almost all of the sites (see Figure 5). For $SO_2$ this is a consequence of the sparsely distributed measurement sites being located in rural regions remote from any of the sources in USA–W (Clarke et al., 1997; MACTEC-Engineering and Consulting, 2005; Baumgardner et al., 2002) (see also Figure B1). This results in some very large NMB values in USA–W (see Figure 5a).

## 3.4 Seasonal Cycles

Figure 7 shows modelled and observed surface $SO_2$ and $SO_4^{2-}$ concentrations averaged seasonally for the period 1987–2014. The comparison between the model and observations is summarised for DJF and JJA in Table 5. The higher winter time $SO_2$ concentrations are driven by greater emissions from coal fired power plants and domestic heating, and less oxidation. Conversely there are fewer emissions and higher oxidant concentrations in summer time. These cycles drive correspondingly low

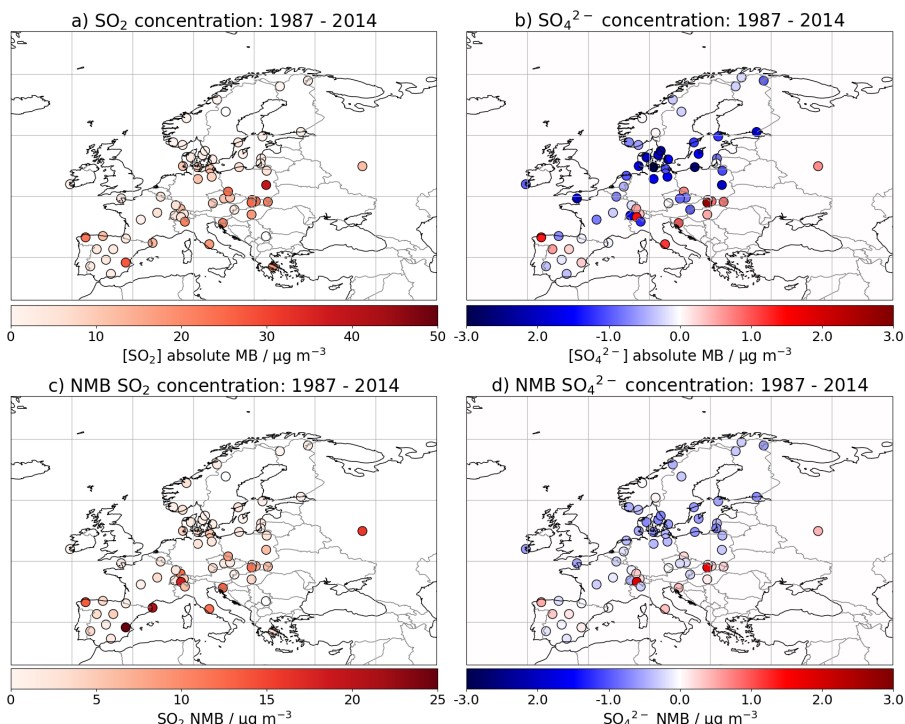

**Figure 6.** Geographic distribution of mean bias (UKESM1 - obs) in mean annual surface $SO_2$ and $SO_4^{2-}$ concentration at EMEP measurement sites ($N$ sites = 48 for $SO_2$ and 42 for $SO_4^{2-}$). The mean annual surface concentrations are calculated over the period 1987 to 2014. Absolute mean bias is shown in (a) and (b) and normalised mean bias (NMB) is shown in (c) and (d). Note that different scales are used for the $SO_2$ model bias (a) and normalised mean bias (c).

$SO_4^{2-}$ concentrations in winter and high $SO_4^{2-}$ concentrations in summer. Overall, we find that the model bias in surface $SO_2$ and $SO_4^{2-}$ concentrations depends on the season as well as the region and pollution levels.

The results presented in Sections 3.1 and 3.2 show that UKESM1 consistently over-predicts mean annual surface $SO_2$ con-
centrations in the USA and Europe, however Figures 7a-c show that in the more polluted regions (USA–E and Europe), the magnitude of the bias is seasonal, although still with a large positive bias. UKESM1 is able to capture the seasonal cycle in surface $SO_2$ concentrations over Europe, but the absolute model bias is larger in DJF compared with JJA (see Table 5). In USA–E UKESM1 does not capture the seasonal cycle in surface $SO_2$ concentrations due to the relatively large model bias in JJA, where the modelled $SO_2$ concentrations are over five times the observed values. In USA–W the modelled and observed
surface $SO_2$ concentrations are slightly higher in DJF compared with JJA, but the model bias is so large in this region that it is difficult to determine if there is any seasonality in this bias.

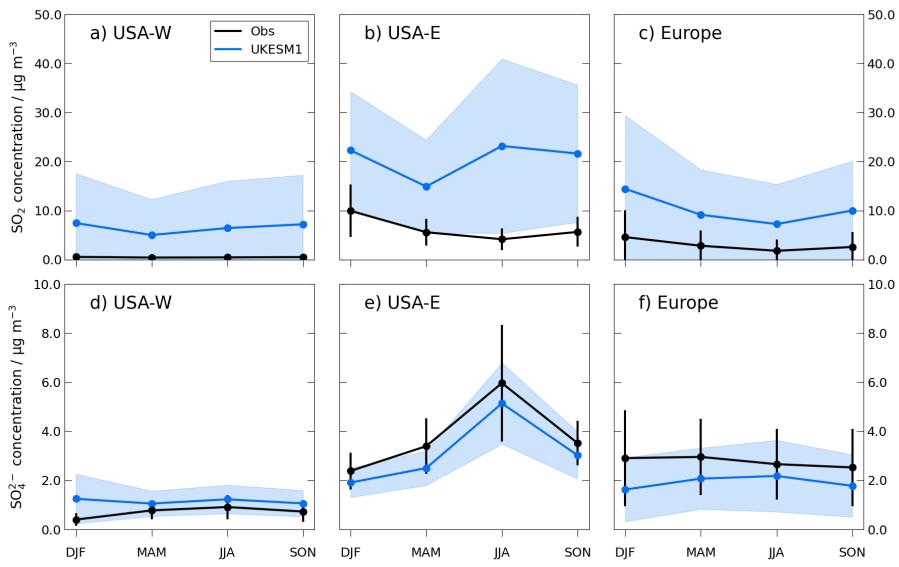

**Figure 7.** Modelled and observed seasonal mean surface $SO_2$ concentration (top row) and surface $SO_4^{2-}$ concentration (bottom row) for the period 1987–2014. USA–W, (a, d; $N$ sites = 16); USA–E, (b, e; $N$ sites = 33) and EMEP (c, e; $N$ sites ($SO_2$) = 48 and $N$ sites ($SO_4^{2-}$) = 42). The blue shaded region and the black error bars represent the standard deviation across the sites in the observational network.

UKESM1 clearly captures the seasonal cycle in surface $SO_4^{2-}$ concentration over USA–E, simulating the highest values in summer and the lowest values in winter. The model under-predicts surface $SO_4^{2-}$ concentration by a factor of 0.7 to 0.8 reasonably consistently throughout the seasonal cycle. In the cleaner USA–W region, UKESM1 is able to capture the seasonal cycle in surface $SO_4^{2-}$ concentrations, with the exception of DJF where the model over-predicts surface $SO_4^{2-}$ concentrations by a factor of 2.5. In Europe the observed seasonal cycle in surface $SO_4^{2-}$ concentration has only a small amplitude with mean values of 2.81 $\mu g\,m^{-3}$ and 2.85 $\mu g\,m^{-3}$ in DJF and JJA respectively.

### 3.5    Evaluation of total column $SO_2$ in UKESM1 against satellite observations

Figure 8 shows total column $SO_2$ (TCSO$_2$) from UKESM1 and OMI, and the difference between them for DJF and JJA. Note that the quality control for solar zenith angle results in no data availability above 65°N degrees or below 65°S in the winter months, and due to OMI's weaker sensitivity to retrieving $SO_2$ in remote regions we focus on comparing TCSO$_2$ over source and outflow regions (Li et al., 2020). Figures 8 a-d show that UKESM1 and OMI broadly agree on the location of the main northern hemisphere (NH) source regions including China, India, Europe and the USA. The model and satellite data both show seasonal cycles in TCSO$_2$ over the large source regions with higher values being modelled and observed during the winter months. However, Figures 8e and f show that the UKESM1 TCSO$_2$ values were generally larger than the OMI TCSO$_2$ values

**Table 5.** Statistics for seasonal mean surface $SO_2$ and $SO_4^{2-}$ concentrations ($\mu g\,m^{-3}$) at USA–W, USA–E and Europe. The mean seasonal values are averaged over the period 1987–2014.

| | USA–W | | USA–E | | Europe | |
|---|---|---|---|---|---|---|
| | DJF | JJA | DJF | JJA | DJF | JJA |
| $SO_2$ concentration | | | | | | |
| Mean (obs) | 0.71 | 0.45 | 9.83 | 4.17 | 5.00 | 2.10 |
| Mean (model) | 7.39 | 6.32 | 22.18 | 22.93 | 15.34 | 8.08 |
| Bias | 6.69 | 5.87 | 12.34 | 18.76 | 10.34 | 5.99 |
| NMB | 12.16 | 12.85 | 1.37 | 5.01 | 2.64 | 3.09 |
| N sites | 16 | 16 | 33 | 33 | 48 | 48 |
| $SO_4^{2-}$ concentration | | | | | | |
| Mean (obs) | 0.48 | 0.95 | 2.71 | 6.51 | 2.81 | 2.85 |
| Mean (model) | 1.23 | 1.21 | 1.88 | 5.15 | 1.72 | 2.50 |
| Bias | 0.75 | 0.25 | -0.83 | -1.35 | -1.09 | -0.35 |
| NMB | 1.63 | 0.26 | -0.32 | -0.22 | -0.38 | -0.13 |
| N sites | 16 | 16 | 33 | 33 | 42 | 42 |

**Table 6.** Statistics comparing model and OMI Total Column $SO_2$ over three regions; USA ($60-130°W$, $25-50°N$), Europe ($15°W-40°E$, $35-65°N$) and South to North East Asia ($75-125°W$, $20-45°N$) for the period 2005-2014. The metrics are mean bias (MB, DU), root mean square error (RMSE, DU), percentage mean bias (MB%) and correlation (R).

| | USA | | Europe | | South to North East Asia | |
|---|---|---|---|---|---|---|
| Statistic | UKESM1 | UKESM1-SO2 | UKESM1 | UKESM1-SO2 | UKESM1 | UKESM1-SO2 |
| MB | 0.029 | 0.014 | 0.050 | 0.027 | 0.120 | 0.068 |
| RMSE | 0.032 | 0.018 | 0.061 | 0.040 | 0.122 | 0.070 |
| MB% | 117 | 56 | 110 | 59 | 402 | 228 |
| R | 0.26 | 0.19 | 0.54 | 0.53 | 0.64 | 0.48 |

Note that the median value is reported for each metric.

in these source regions by $0.6-1.0$ Dobson Units (DU). Over the background regions UKESM1 over-predicts $TCSO_2$ values by $0.2-0.5$ DU. UKESM1 also has larger volcanic sources and associated outflow, which can be seen over central America, Sicily, Hawaii and Papua New Guinea, for example. This is likely due to the climatology that UKESM1 uses for continuously degassing volcanoes. In agreement with the ground based observations, the satellite data shows an east-west divide in the USA,

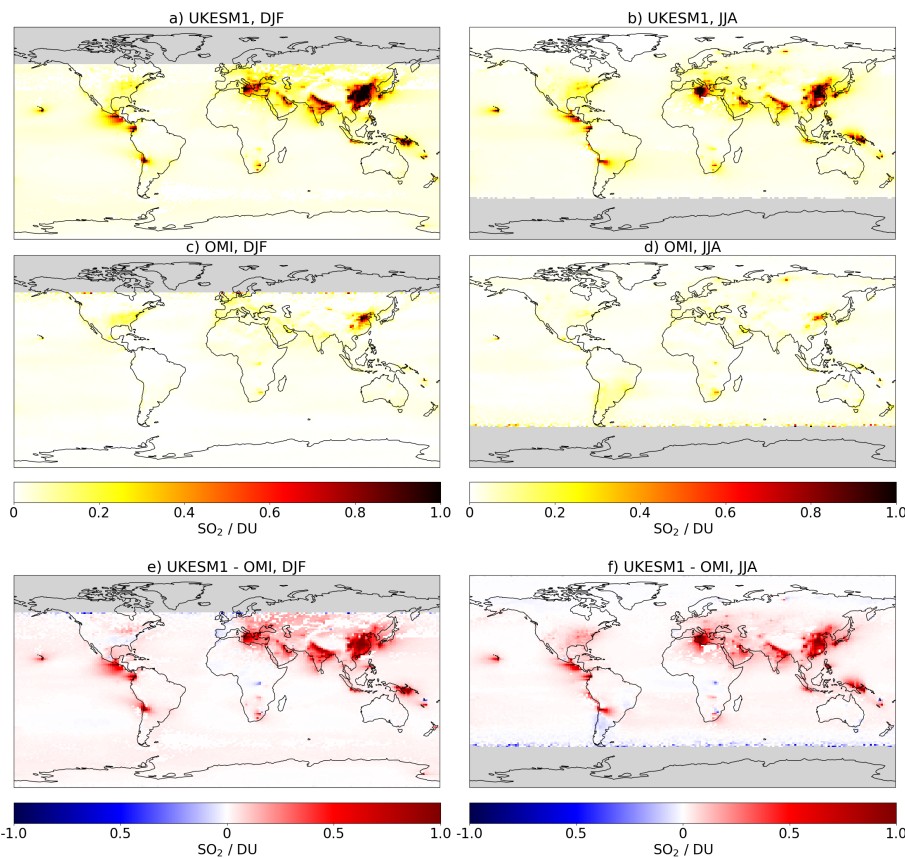

**Figure 8.** Total Column SO$_2$ for UKESM1 (a, b) OMI (c, d) and UKESM1 − OMI (e, f). DJF is shown in the left column and JJA is shown in the right column. Median Total Column SO$_2$ in Dobson Units is calculated for the period 2005–2014.

with greater TCSO$_2$ over USA–E compared with USA–W.

Figure 9 shows modelled and observed TCSO$_2$ over the period from 2005 to 2014 for three source regions, the USA
(60–30°W, 25–50°N), Europe (15°E–40°E, 35–65°N) and South to North East (SNE) Asia (75–125°E, 20–45°N). Overall,
we find that the observed TCSO$_2$ is reasonably stable over this period in all three regions and that there are clear seasonal
cycles showing peak TCSO$_2$ during the NH winter in the USA and Europe, and slightly earlier in SNE Asia. However, Figures
9a and b show that modelled TCSO$_2$ decreases over the USA and Europe from 2005–2010 with UKESM1 over-predicting
TCSO$_2$ by up to 0.1 DU at the start of the time series. After 2011, UKESM1 is in much better agreement with the observed
TCSO$_2$ over both regions. Figure 9c shows that UKESM1 consistently over-predicts TCSO$_2$ over SNE Asia by 1.5–2.0 DU
during the period from 2005 to 2014. UKESM1 does simulate a seasonal cycle in TCSO$_2$ in all three regions. In Europe,
UKESM1 is able to predict the peak winter time TCSO$_2$ values, although between 2005–2010 the model has a positive bias of
up to 0.1 DU. However, in the USA and to a lesser extent in SNE Asia (Figures 9a and c respectively), UKESM1 mis-times the

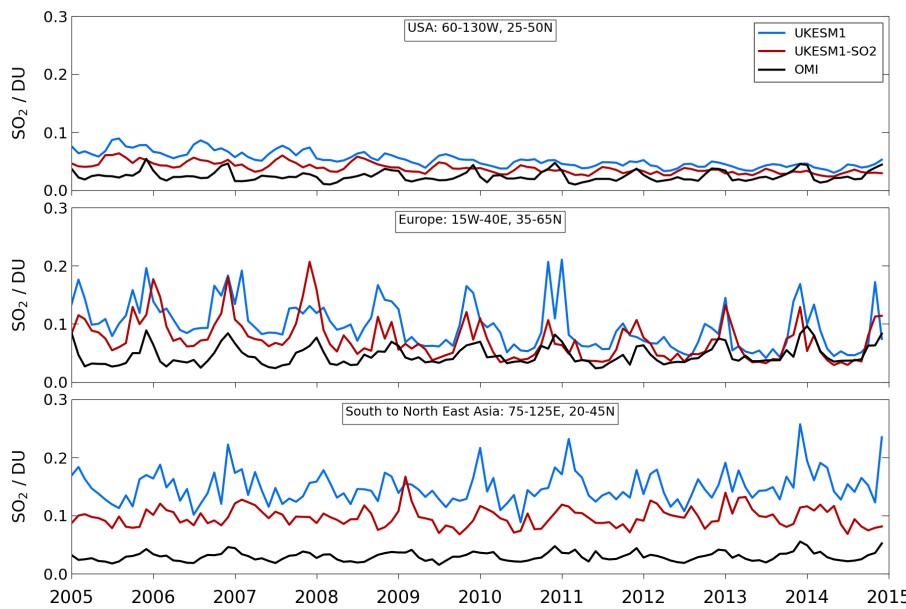

**Figure 9.** Median Total Column $SO_2$ calculated for 2005–2014 for USA, (a); Europe, (b); and South to North East Asia, (c). Total Column $SO_2$ is in Dobson Units.

peak $TCSO_2$ values. In the USA the model simulates the highest values in the summer rather than in winter, and in SNE Asia
the modelled peak $TCSO_2$ values appear shifted several months earlier relative to the observations.

The observations of $TCSO_2$ and surface $SO_2$ concentrations over Europe both show the impact of emission control policies
on keeping atmospheric $SO_2$ levels low (Figures 2c and 9b). However, for the USA, we investigate the surface $SO_2$ concentrations separately over the USA–E and USA–W regions, whereas the $TCSO_2$ is averaged over continental USA as a whole.
As a result $TCSO_2$ does not appear to decrease over the period 2005–2014 in the same way surface $SO_2$ concentrations are
reduced in USA–E (Figures 2b and 9a). Using a mean $TCSO_2$ for the continental USA as a whole also results in lower values
relative to Europe. This is in contrast to the ground-based observations where surface $SO_2$ concentrations over Europe are
intermediate between USA–E and USA–W for the period 2005–2014. The relatively high $TCSO_2$ over Europe is also due to
the inclusion of a number of large eastern European sources which are not well represented in the ground based observations.
We find that the $TCSO_2$ and surface $SO_2$ concentration observations both agree on the seasonal cycle, showing higher values
in winter compared with the summer.

UKESM1 over-predicts surface $SO_2$ concentration to a greater extent than $TCSO_2$ for the USA and Europe. The model
over-predicts surface $SO_2$ concentration by a factor of 2.2 - 11.6 for USA–E and USA–W, and 2.4 for Europe compared with
values of 1.2 (USA) and 1.1 (Europe) for modelled $TCSO_2$ (see Tables 3 and 6). However, the R values for surface $SO_2$

concentration (>0.8) are much better than those for TCSO$_2$ (0.26 - 0.53), particularly over the USA. The low R value for the USA reflects the poor seasonal agreement in TCSO$_2$ in this region. The comparison against both observational data sets shows that the modelled atmospheric SO$_2$ is too high, both at the surface and through the column. In addition, UKESM1 simulates larger trends in TCSO$_2$ and surface SO$_2$, particularly prior to 2010, than are seen in the observations. Both observational data sets also show that UKESM1's over-prediction of atmospheric SO$_2$ in the large source regions is generally greater in the winter months compared to the summer months (see Figures 8e and f). Exceptions occur in USA–E where UKESM1 fails to capture the seasonal cycle in atmospheric SO$_2$, over predicting surface SO$_2$ concentration and TCSO$_2$ to a greater extent in JJA compared to DJF. Notably, in the southern USA–E region and the Iberian peninsula UKESM1 actually under-predicts TCSO$_2$ by up to $0.1\,\mathrm{DU}$ in DJF, which does not occur in the comparison with surface SO$_2$ concentrations, even if the model and observations are compared at individual CASTNet and EMEP sites.

## 4 Impact of changes to the SO$_2$ dry deposition parameterization in UKESM1

### 4.1 Global scale impacts

Figure 10 shows SO$_2$ dry deposition velocity simulated by UKESM1 and how this is affected by changes to the SO$_2$ dry deposition parameterization. In UKESM1 the mean annual deposition velocities range from approximately $0.5\times10^{-3}$ to $4.0\times10^{-3}$ $\mathrm{m\,s^{-1}}$ (see Figure 10a). Figures 10b, e, and h show that SO$_2$ dry deposition velocity increases over almost all land and ocean regions in UKESM1-SO2 by approximately $0.02$ - $0.08\,\mathrm{m}\,s^{-1}$. This represents an increase of a factor of 2 - 4 relative to UKESM1. The largest increase in SO$_2$ dry deposition velocity to the ocean is over the Southern Ocean in winter where it increases by more than a factor of four (see Figure 10i). Over land surfaces the largest increases occur over South America, north western America and Canada, and north eastern Europe/western Russia (see Figures 10f and i). The increased SO$_2$ dry deposition velocities in UKESM1-SO2 relative to UKESM1 indicate that the changes to the SO$_2$ dry deposition parameterization are behaving as expected. The reduction in $R_c$ for SO$_2$ dry deposition to water increases the dry deposition velocity to oceans. Similarly, when land surfaces are allowed to remain wet for a longer period after rainfall events, $R_{cut}$ and $R_{soil}$ are reduced for a longer period of time in UKESM1-SO2 relative to UKESM1 and SO$_2$ dry deposition velocity to the canopy (leaf or soil) increases. In the summer months SO$_2$ dry deposition velocities are larger over land surfaces compared with in the winter months, while over oceans values are larger in the winter compared with the summer. Although increased rainfall in winter drives wetter surface conditions, the leaf canopy is reduced or absent, and at high latitudes surfaces are likely to have snow cover which has a relatively high $R_c$ (see Table A1) compared with vegetated surfaces. The higher SO$_2$ dry deposition velocities over the ocean during the winter months are likely due to higher wind speeds.

The increased SO$_2$ dry deposition velocities in UKESM1-SO2 drive an increase in the SO$_2$ dry deposition flux of nearly 45% (from $29.49\,\mathrm{T\,g}\,y^{-1}$ to $42.56\,\mathrm{T\,g}\,y^{-1}$) and subsequently reduce the SO$_2$ lifetime by approximately 25% (see Table C1). Overall the global SO$_2$ burden is reduced from $0.54\,\mathrm{T\,g}\,y^{-1}$ to $0.41\,\mathrm{T\,g}\,y^{-1}$. The full budget breakdown for SO$_2$ is given in

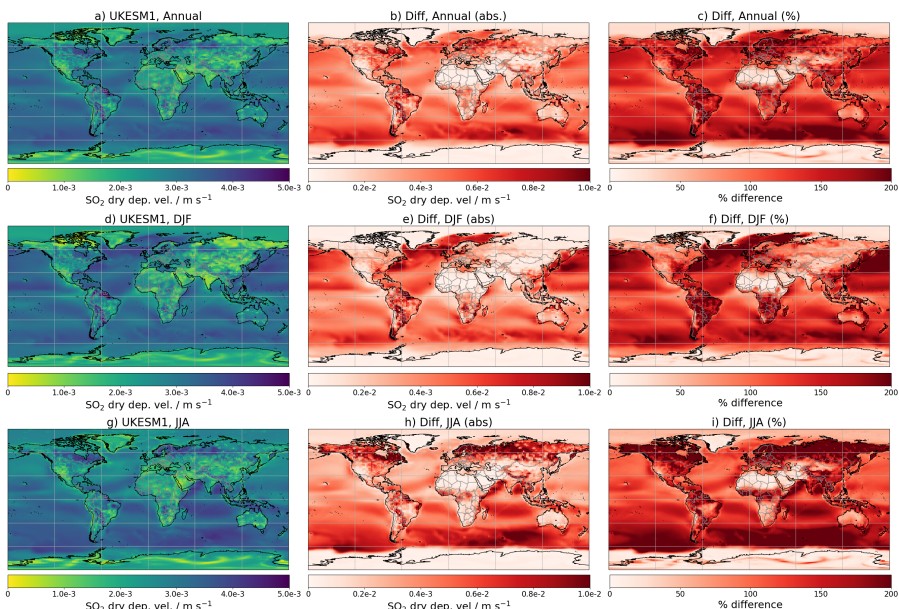

**Figure 10.** $SO_2$ dry deposition velocity for UKESM1, (left column); absolute difference in $SO_2$ dry deposition velocity for UKESM1-SO2 - UKESM1, (middle column) and percentage difference in $SO_2$ dry deposition velocity for UKESM1-SO2 - UKESM1, (right column). Annual means are shown in a–c, the DJF mean is hown in d–f and the JJA mean is shown in g–i). Each plot shows mean data for the period 1987–2014. Deposition velocity is calculated from the simulated surface $SO_2$ and dry deposition flux values and represents the 'bulk' deposition velocity for each model grid cell.

Appendix C (Table C1). The spatial distribution of these changes is illustrated in Figure 11 which shows that $SO_2$ dry depo-
sition increases over most regions, with a corresponding decrease in surface $SO_2$ concentrations. The lower $SO_2$ burden then drives lower oxidation fluxes (Table C1), reducing the surface $SO_4^{2-}$ concentrations (Figure 11g-i).

The largest absolute increases in $SO_2$ dry deposition occur over the main source regions of USA–E, eastern China, and central and eastern Europe (Figure 11b). However, Figure 11c shows that the largest relative changes in dry deposition are over
the ocean. Although the absolute changes over the ocean are small ($<1 \times 10^{-5}\,\mathrm{kg\,m^{-2}\,y^{-1}}$ ), the large global surface area of ocean means that this increase is important for the global sulphur cycle. In UKESM1-SO2 $R_{surf}$-mod increases the length of time land surfaces remain wet after rainfall, thus increasing $SO_2$ dry deposition over most land surfaces (as seen in Figures 11b) due to the high solubility of $SO_2$ in water. Note that $R_{surf}$-mod also makes $SO_2$ dry deposition a function of the soil moisture content and this aspect of the change drives decreases in $SO_2$ dry deposition of 20-40% over desert regions, such as
the Sahara and high northern latitudes (see Figures 11b and c). Conversely $R_{surf}$-mod does not impact ocean surfaces and the increases in $SO_2$ dry deposition over oceans are driven by $R_{water}$-mod.

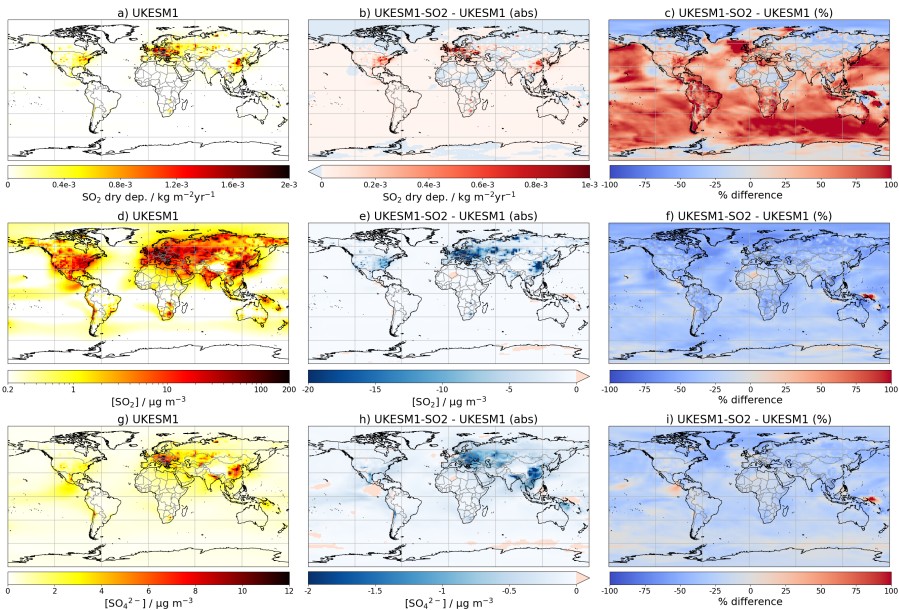

**Figure 11.** $SO_2$ dry deposition, (a–c); surface $SO_2$ concentration, (d–f) and surface $SO_4^{2-}$ concentration (g–i) for UKESM1 (left column), absolute difference for UKESM1-SO2 - UKESM1 (middle column) and percentage difference for UKESM1-SO2 - UKESM1 (right column). In the plots showing absolute difference, areas where $SO_2$ dry deposition is reduced in UKESM1-SO2 compared with UKESM1 are shown in blue (b) and areas where surface $SO_2$ and $SO_4^{2-}$ concentration are greater in UKESM1-SO2 compared with UKESM1 are shown in red (e, h). The blue and red shading in (b) and (e, h) is not to scale as the absolute differences are very small. Each plot shows mean data for the period 1987–2014.

The largest absolute reductions in mean annual surface $SO_2$ and $SO_4^{2-}$ concentrations are over the source regions, corresponding with the locations of the largest increases in $SO_2$ dry deposition. This is expected because dry deposition of $SO_2$ to
the surface is directly proportional to the surface concentration (Equation 1). Figure 11e shows that mean annual surface $SO_2$ concentration was reduced by up to $20\,\mu g\,m^{-3}$ in the eastern USA, eastern China, and central and eastern Europe, which corresponds to a percentage decrease of 30-50%. Note that this is similar to the percentage decrease in mean annual surface $SO_2$ concentration over remote and ocean regions, although the absolute fluxes are much larger over the source regions. With the exception of some areas in the Sahara and the Middle East, we do not see increases in mean annual surface $SO_2$ concentration
where dry deposition fluxes decrease (albeit by very low amounts), such as the Arctic. We suggest that this is because these areas are remote and contain no $SO_2$ sources and by reducing $SO_2$ in the source regions, we reduce overall atmospheric $SO_2$ loading and therefore less is transported to remote areas. Figures 11h and i show that mean annual surface $SO_4^{2-}$ concentrations are also reduced over the main source regions, although the reductions over the USA are relatively small compared to the other large source regions ($0.5\,\mu g\,m^{-3}$ compared with up to $3\,\mu g\,m^{-3}$ in central and eastern Europe and China). Proportionally, the
reduction in mean annual surface $SO_4^{2-}$ concentration is smaller than that for mean annual $SO_2$ concentration, with decreases

generally less than 5% over most source regions.

## 4.2 Evaluation of UKESM1-SO2 against ground-based observations

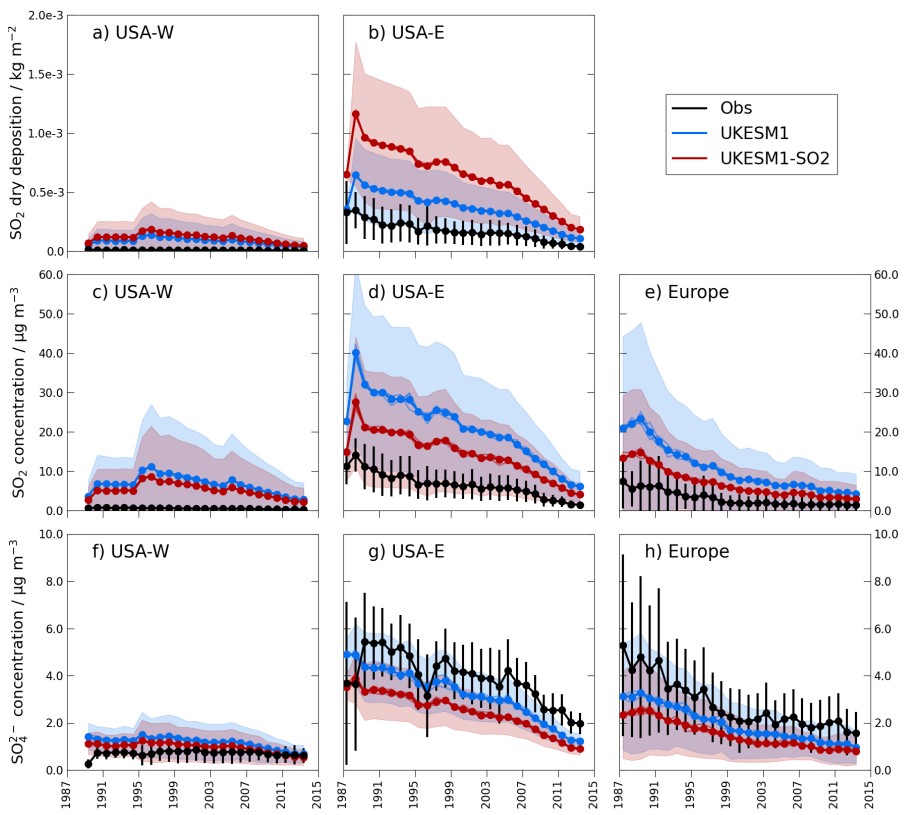

**Figure 12.** Time series of observed and modelled mean annual $SO_2$ dry deposition flux (top row), surface $SO_2$ concentration (middle row) and $SO_4^{2-}$ concentration (bottom row) for USA–W, (a, c, f; $N$ sites = 16); USA–E, (b, d, g; $N$ sites = 33) and Europe (e, h; $N$ $SO_2$ sites = 48 and $N$ $SO_4^{2-}$ sites = 42). No $SO_2$ dry deposition flux observations are available for Europe.

In Figure 12 and Table 7 we evaluate UKESM1-SO2 against the ground-based observations of mean annual surface $SO_2$ 575 and $SO_4^{2-}$ concentration for the USA and European regions over the period from 1987–2014. UKESM1-SO2 is also evaluated against $SO_2$ dry deposition flux from CASTNet over the same period. Figures 12 a and b show that $SO_2$ dry deposition flux is increased in UKESM1-SO2 relative to UKESM1 with this increase being more pronounced over USA–E compared with USA–W (see also Table 7). The increase in $SO_2$ dry deposition in UKESM1-SO2 does enhance the model's over-prediction of this parameter relative to the CASTNet data, with NMB increasing from 1.15 in UKESM1 to 3.0 in UKESM1-SO2. Mean 580 annual $SO_2$ dry deposition fluxes are very low over USA–W due to the much lower concentrations of $SO_2$ in this region.

**Table 7.** Statistics for mean annual surface $SO_2$ and $SO_4^{2-}$ concentrations and $SO_2$ dry deposition flux for UKESM1 and UKESM1-SO2. The mean seasonal values are averaged over the period 1987–2014. The units for $SO_2$ and $SO_4^{2-}$ concentrations are $\mu g\,m^{-3}$ and the units for $SO_2$ dry deposition flux are $kg\,m^{-2}\,y^{-1}$ for $SO_2$.

| | USA–W | | USA–E | | Europe | |
| --- | --- | --- | --- | --- | --- | --- |
| | UKESM1 | UKESM1-SO2 | UKESM1 | UKESM1-SO2 | UKESM1 | UKESM1-SO2 |
| **$SO_2$ dry deposition** | | | | | | |
| Mean (obs) | $6.11\times10^{-6}$ | - | $1.64\times10^{-4}$ | - | - | - |
| Mean | $8.16\times10^{-5}$ | $1.17\times10^{-4}$ | $3.50\times10^{-4}$ | $6.17\times10^{-4}$ | - | - |
| Bias | $7.75\times10^{-5}$ | $1.03\times10^{-4}$ | $1.87\times10^{-4}$ | $4.53\times10^{-4}$ | - | - |
| NMB | 12.41 | 16.96 | 1.27 | 2.98 | - | - |
| N sites | 16 | - | 33 | - | - | - |
| **$SO_2$ concentration** | | | | | | |
| Mean (obs) | 0.48 | - | 6.34 | - | 2.94 | - |
| Mean (model) | 6.48 | 5.02 | 20.50 | 14.10 | 10.20 | 6.60 |
| Bias | 6.00 | 4.54 | 14.20 | 7.72 | 7.26 | 3.67 |
| NMB | 12.54 | 9.52 | 2.41 | 1.34 | 2.61 | 1.36 |
| N sites | 16 | - | 33 | - | 48 | - |
| **$SO_4^{2-}$ concentration** | | | | | | |
| Mean (obs) | 0.70 | - | 3.82 | - | 2.76 | - |
| Mean | 1.15 | 0.95 | 3.14 | 2.42 | 1.91 | 1.48 |
| Bias | 0.43 | 0.24 | -0.67 | -1.39 | -0.85 | -1.28 |
| NMB | 0.72 | 0.41 | -0.19 | -0.38 | -0.31 | -0.47 |
| N sites | 16 | - | 33 | - | 42 | - |

UKESM1 does over-predict mean annual $SO_2$ dry deposition flux in this region too, but the absolute bias changes very little in UKESM1-SO2 (see Table 7). Figures 12 c - e show that model bias in mean annual surface $SO_2$ concentration is reduced in UKESM1-SO2 compared with UKESM1 in all three regions over the period 1987–2014. The largest absolute reduction is in USA–E where mean annual surface $SO_2$ concentration decreases from 20.05 $\mu g\,m^{-3}$ to 14.10 $\mu g\,m^{-3}$, however, the largest reduction in NMB is in USA–W where it decreases from 12.54 to 9.52 (see Table 7). The model's over prediction of mean annual surface $SO_4^{2-}$ concentration is reduced over USA–W in UKESM1-SO2 compared with UKESM1, with NMB decreasing from 0.72 to 0.41. However, the model's under prediction of mean annual surface $SO_4^{2-}$ concentration over USA–E and Europe increases, with NMB = -0.31 in UKESM1 and NMB = -0.47 in UKESM1-SO2 (see Table 7). We find that the changes to the surface concentration and dry deposition flux occur almost uniformly over the seasonal cycle and so do not change the

 patterns in seasonal bias that are described in Section 3.4 for surface $SO_2$ and $SO_4^{2-}$ concentrations.

## 4.3 Evaluation of UKESM1-SO2 against TCSO$_2$ observations

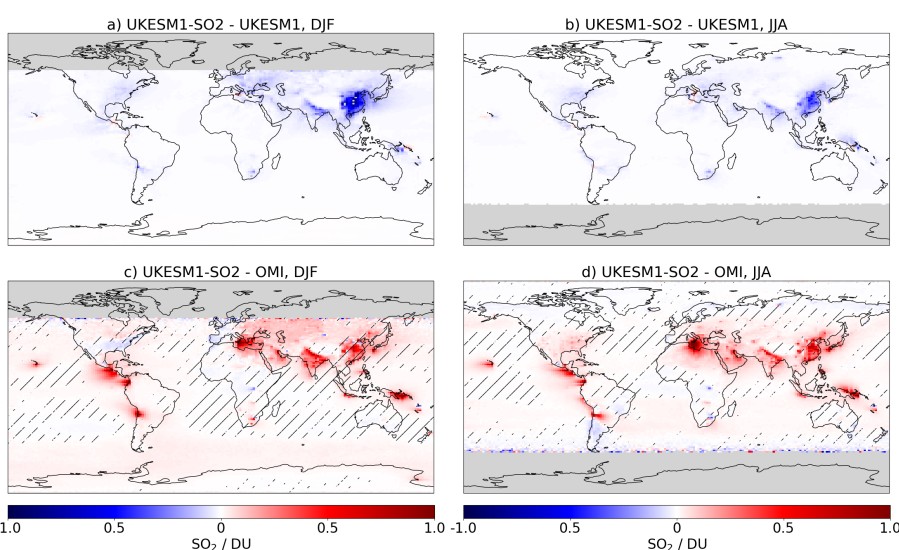

**Figure 13.** Difference in TCSO$_2$ between UKESM1 and UKESM-SO2 for DJF (a) and JJA (b). Difference in TCSO$_2$ between UKESM1-SO2 and OMI for DJF (c) and JJA (d). Model and satellite data is averaged over the period 2005–2014. The hatched regions show where the inter-model differences are smaller than the existing model-satellite difference

The impact of the modifications to the $SO_2$ dry deposition parameterization on TCSO$_2$ are shown in Figure 13 and Table 6. Figures 13 a and b show that TCSO$_2$ over the source regions is lower in UKESM1-SO2 relative to UKESM1 by $0.1 - 0.5$ DU in DJF and JJA. This results in UKESM1-SO2 having a smaller positive bias in TCSO$_2$ of $+0.3 - +0.5$ DU compared with that of $+0.6 - +1.0$ DU for UKESM1, (Figures 13c and d; Section 3.5 and Figure 8). This represents a decrease in the global TCSO$_2$ model - OMI bias of 20-30%. Over the outflow regions (e.g. off the USA eastern seaboard), TCSO$_2$ has reduced by 30-50% and over the source regions, this varies by 30-50% for South to North East Asia, 20-30% for Europe and 10-30% for the USA (see Table 6). However, Figures 13c and d also show that the inter-model differences are smaller than the existing model-satellite difference, i.e. the hatched regions are sporadic with limited coverage.

## 5 Discussion

The evaluation of UKESM1 against ground-based observations of $SO_2$ and $SO_4^{2-}$ concentrations from the USA and Europe, as well as $SO_2$ dry deposition fluxes from the USA, shows that the model is able to represent recent historical changes in

these variables. UKESM1 is also able to capture the spatial patterns in surface $SO_2$ and $SO_4^{2-}$ concentrations and $SO_2$ dry deposition, simulating larger values close to the sources and lower values away from the sources. However, UKESM1 generally over-predicts surface $SO_2$ concentrations and dry deposition fluxes, while under-predicting surface $SO_4^{2-}$ concentrations for the period 1987-2014. Further, we find that UKESM1 over-predicts the rate at which the surface $SO_2$ concentrations decrease over this period.

We also make use of the updated $TCSO_2$ product from OMI to evaluate UKESM1, finding that the model captures spatial patterns in $TCSO_2$ at the global scale. Importantly, this evaluation allows us to identify model bias in regions without long-term ground-based networks, showing that UKESM1 over-predicts $TCSO_2$ over all source and outflow regions. We find that although the ground-based and satellite observations are subject to different uncertainties, UKESM1's relative over-prediction of both surface $SO_2$ concentration and $TCSO_2$ is similar in the USA and Europe. This suggests that our finding of positive bias in modelled atmospheric $SO_2$ is robust. We have also demonstrated that a more realistic treatment of $SO_2$ dry deposition in UKESM1 reduces the model's atmospheric loading of $SO_2$ and $SO_4^{2-}$. However, we find that UKESM1's under-prediction of surface $SO_4^{2-}$ concentrations and over-prediction of $SO_2$ dry deposition fluxes increases when the changes are included in the model, suggesting that there are further uncertainties UKESM1's representation of the complex sulphur cycle processes. Additionally, the spatial and temporal differences in the model bias suggests that the drivers of model bias are regionally and seasonally dependant.

Broadly, a model's over prediction of atmospheric $SO_2$ can be driven by too little removal of $SO_2$ (via deposition or oxidation) or too high emissions. UKESM1 uses $SO_2$ emissions from CMIP6 (Eyring et al., 2016). In comparing these emissions with the HTAP-OMI (Liu et al., 2018) and EDGAR (Crippa et al., 2018) data sets, Pope and Chipperfield (2021) showed that the total $SO_2$ emissions in CMIP6 (115 Tg y$^{-1}$) are moderately larger than the HTAP-OMI (100 Tg y$^{-1}$) and EDGAR data sets (102 Tg y$^{-1}$). In general the CMIP6 emissions are larger than the HTAP-OMI and EDGAR emissions over all major source regions by up to $1\times10^{-10}$ kg m$^{-2}$ y$^{-1}$. Exceptions occur at a number of point locations, which are likely from OMI rather than HTAP (Liu et al., 2018). Additionally, Smith (2021) has reported that $SO_2$ emissions over the western USA are too high, possibly because the proxy data used to spatially distribute emissions does not take into account lower sulphur coal used in power plants and industry in this region.

Emission injection height is also an important constraint on near surface $SO_2$ concentrations as demonstrated by Yang et al. (2019). This study found that uncertainty in industrial emission height resulted in modeled near-surface $SO_2$ concentrations varying between 70% and 130% over most land regions, higher than the overall uncertainty of 8-14% attributed to $SO_2$ emission rates. The $SO_2$ injection height in UKESM1 was investigated by Mulcahy et al. (2020) who used injection heights prescribed as for HadGEM-GC3.1, where 50% of energy and industry sector emissions are injected in to the atmosphere at a height of 500 m. Mulcahy et al. (2020) showed that the introduction of a vertical profile for $SO_2$ emissions in UKESM1 had negligible impact on surface $SO_4^{2-}$ concentrations at measurement sites in Europe and the USA, suggesting an important role

for the aerosol chemistry in these regions. Emitting the $SO_2$ at higher altitudes will act to reduce surface $SO_2$ concentrations and therefore model's bias against surface observations of both $SO_2$ concentration and $SO_2$ dry deposition flux. However, Pope and Chipperfield (2021) showed that UKESM1's bias in $TCSO_2$ increased when emission injection heights were increased. This also suggests that the CMIP6 emissions are too high and that using a vertical profile for the emissions to some extent shifts the model's bias in $SO_2$ to higher altitudes. However, we note that using varying emission heights for $SO_2$ did not affect

column densities in the GEOS-5/GOCART model (Buchard et al., 2014). We suggest that undertaking model experiments with different emissions inventories and injection height profiles to cast light on the role of $SO_2$ emissions in model bias in the sulphur cycle.

The two main removal pathways for $SO_2$ are oxidation to sulphate and dry deposition to the Earth's surface. In this study

we have evaluated a more realistic treatment of $SO_2$ dry deposition in UKESM1 that accounts for the high solubility of $SO_2$ in water. We find that this reduces the dry deposition lifetime and consequently reduces the overall $SO_2$ burden and lifetime. This reduces positive model bias in surface $SO_2$ concentrations in the USA and Europe, and in $TCSO_2$ across most of the globe. The changes have the largest impact over source regions because dry deposition flux is directly proportional to the atmospheric concentration of $SO_2$. There is also a reduction in the $SO_2$ oxidation lifetime which likely drives the reduced atmospheric

loading of $SO_4^{2-}$. However, the true impact of the changes to the dry deposition parameterization are confounded by model uncertainty in other aspects of the complex sulphur cycle as well as the inherent difficulties associated with evaluating a global model against point observations.

Dry deposition is a highly parameterized process and often poorly represented, particularly in global models. Similarly to

UKESM1, Vet et al. (2014) showed that $SO_2$ dry deposition fluxes were over-predicted by the 23 member model ensemble used for the TF-HTAP exercise relative to inferential data sets from measurement networks (including CASTNet). A key uncertainty highlighted in this study was that associated with the inferred dry deposition fluxes; from CASTNet these could be up to 30% lower than direct observations of $SO_2$ dry deposition flux (Baumgardner et al., 2002). However, the fluxes simulated by UKESM1 are a factor of 2 to 10 higher than the inferred data from USA–E and USA–W, indicating that the modelled

deposition fluxes are almost certainly too large. Additional sources of uncertainty in model simulations of $SO_2$ dry deposition may include land surface cover, changes in the atmospheric $SO_2:NH_3$ ratio and the ratio between wet and dry deposition of $SO_2$. Mulcahy et al. (2020) showed that wet deposition, and to a lesser extent dry deposition, of $SO_2$ was considerably lower in UKESM1 compared with HadGEM-GC3.1. Paulot et al. (2017) also report that poor representation of wet deposition likely contributed to bias in modelled surface $SO_4^{2-}$ concentrations. Too low wet deposition would also contribute to the model's

over-prediction of surface $SO_2$ concentration. Dry deposition is sensitive to land surface type, which may not be well captured in global models. In this study the UKESM1 configuration uses 13 land cover classes including 11 plant functional types (Archibald et al., 2020). This is reasonable for a global model, but inevitably detail is lost. Vet et al. (2014) also suggest that $SO_2$ dry deposition may depend on the atmospheric $NH_3$ loading. Long term measurements at a UK site showed that $SO_2$ dry deposition velocity has increased with time, which was attributed to changing ratios of $SO_2:NH_3$ as $SO_2$ concentrations have

decreased faster than $NH_3$ concentrations (Vet et al., 2014; Fowler et al., 2009). Currently nitrate chemistry is not represented in UKESM1, although it is planned for future model versions, and $NH_3$ has not been evaluated in the model, so it is unknown how these factors may contribute to the model's bias in $SO_2$ dry deposition flux and $SO_2$ concentrations.

The role of uncertainty in sulphur cycle chemistry becomes apparent when we consider UKESM1's bias in surface $SO_4^{2-}$
concentrations in combination with the biases in $SO_2$ concentrations and dry deposition. In Europe and USA–E, UKESM1 under-predicts surface $SO_4^{2-}$ concentrations, despite the large positive biases in $SO_2$ concentrations through much of the period from 1987-2014. Note that there are exceptions close to certain point sources, particularly in Europe. However, in the cleaner USA–W region surface $SO_4^{2-}$ concentrations are consistently over-predicted. We suggest that in USA–W too high $SO_2$ emissions (Smith, 2021), possibly in combination with too low emission heights and the associated biases in dry deposition, drive
the model's over-prediction of surface $SO_4^{2-}$ concentrations in this region. However, in the polluted regions of Europe and USA–E the under-prediction of surface $SO_4^{2-}$ concentrations, despite large over-predictions of surface $SO_2$ concentrations suggests that sulphur cycle chemistry is not correctly represented. And Mulcahy et al. (2020) showed that there are global and regional differences in oxidant concentrations and in the $SO_2$ lifetime between UKESM1 and the HadGEM-GC3.1 model, with the latter better able to capture surface $SO_4^{2-}$ concentrations. In reducing the $SO_2$ burden we further reduce its overall
lifetime and oxidation lifetime relative to HadGEM-GC3.1. This highlights the requirement for a more detailed investigation of $SO_2$ oxidation in UKESM1, particularly in polluted regions.

Uncertainty in UKESM1's sulphur chemistry also appears to be seasonally dependant. In USA–E and Europe UKESM1 over-predicts $SO_2$ (surface concentration and $TCSO_2$) and under-predicts surface $SO_4^{2-}$ at all times of the year, but there is
seasonal variation in these biases. In Europe, the model bias is largest in DJF which drives a stronger seasonal cycle relative to the observations. This is in agreement with Turnock et al. (2015) who investigated $SO_4^{2-}$ in an earlier HadGEM3-UKCA configuration. The seasonality in UKESM1's bias over Europe may be due to the model under-predicting in-cloud $SO_2$ oxidation via $O_3$ (Table 1). In winter and at higher latitudes this reaction is likely to be the dominant removal pathway due to lower availability of $H_2O_2$ (Turnock et al., 2019). However, the picture is different in USA–E, where UKESM1 over-predicts $SO_2$
to a larger extent in JJA compared with DJF and there is little seasonality in the model's under-prediction $SO_4^{2-}$. In USA–E it is likely that uncertainty in UKESM1's representation of $SO_2$ oxidation via both $O_3$ and $H_2O_2$ (Table 1) contribute to bias in $SO_2$ and $SO_4^{2-}$. The lower average latitude of the USA–E sites compared with European sites, means that the $O_3$ oxidation pathway is more important in this region and it has been demonstrated that $SO_4^{2-}$ concentrations are sensitive to both pathways in winter Paulot et al. (2017). From this study it is not clear what is driving the relatively large positive bias in $SO_2$ in JJA over
USA–E and we stress the need for closer examination of the sulphur chemistry in UKESM1.

Recent studies have also shown that cloud water pH may be an important factor in the aqueous phase oxidation of $SO_2$ to $SO_4^{2-}$ (Turnock et al., 2019). While a temporally and spatially uniform cloud pH of 5 is currently used in UKESM1, observations of this quantity show that it varies in space and time. Observations at an American site showed that mean cloud pH

increased from 4 to 4.8 between 1994 and 2014 (Schwab et al., 2016), cloud pH measured Mt Tai in North China from 2007-2008 were between 3.56 and 7.64, and measurement campaigns between 1985 and 2008 at various European, North American and East Asian locations reported values between 3.34 and 5.29 (Guo et al., 2012). Turnock et al. (2019) showed that varying this value in the HadGEM3-UKCA model can have a large impact on $SO_2$ and $SO_4^{2-}$ concentrations. Over source regions, including Europe and North America, increasing the cloud water pH by 1.0 reduced the annual mean global $SO_2$ column burden by approximately 50% as more $SO_2$ was oxidized in cloud droplets, and consequently there were small increases in the annual mean sulphate column burden over these regions. Conversely, outside of polluted regions increasing the cloud water pH reduced the sulphate column burden by 10% to 40% globally. These results indicate that having a more realistic treatment of cloud water pH could reduce UKESM1's biases in $TCSO_2$, and potentially in $SO_2$ and $SO_4^{2-}$ concentrations remote from source regions. However, it is unlikely to be a dominant removal pathway at the surface and any impact on surface $SO_2$ concentrations, especially close to sources, would likely be minimal.

In an Earth system model such as UKESM1, there are inevitably some compromises in the complexity of the chemistry and aerosol scheme as these are computationally expensive. While the sulphur chemistry represented in the UKCA-StratTrop model used in UKESM1 accounts for important $SO_2$ and DMS oxidation reactions, as well as simulating oxidants (rather than the offline oxidant scheme used in HadGEM-GC3.1) it cannot be complete. In the recently developed CRI-Strat scheme the sulphur chemistry reactions are as for UKCA-StratTrop, but there is a more comprehensive treatment of non-methane volatile compounds (NMVOC) (Archer-Nicholls et al., 2020) resulting in higher surface ozone concentrations, particularly over polluted areas in summer, compared with UKCA-StratTrop. As demonstrated by Mulcahy et al. (2020) the increased oxidants in UKESM1 relative to HadGEM-GC3.1 likely contribute to reducing the $SO_2$ lifetime from 4.29 to 3.86 days. CRI-Strat is compatible with UKESM1 and the higher oxidant loading may reduce $SO_2$ oxidation lifetime further with a concurrent increase in $SO_4^{2-}$. Model bias in remote ocean regions may also result from the necessarily simplified DMS oxidation chemistry in UKESM1. A more detailed representation of DMS chemistry over the Southern Ocean was investigated by Revell et al. (2019) who found that surface $SO_2$ concentrations increased over the Southern Ocean, possibly due to including reactions between DMS and halogen species, while $SO_4^{2-}$ concentrations decreased, likely as a result of there being more DMS oxidation reactions.

Model resolution is also likely to be an important source of model bias in this study. In evaluating UKESM1 against the CASTNet and EMEP data sets we are comparing a simulated value generated from a model grid box at a scale of $\approx$ 200-300 km with a point observation. This may be a particular problem for the surface $SO_2$ concentrations and $SO_2$ dry deposition fluxes because in reality a large fraction (20-40%) of $SO_2$ emitted from point sources is lost in the first 100 km, which is sub-grid scale relative to the model grid boxes (Smith and Jeffrey, 1975; Wys et al., 1978). In UKESM1 the sub-grid scale loss is not captured because $SO_2$ is evenly emitted across the grid box and deposition is subsequently calculated. Potentially this drives overall large model biases compared with ground based observations that are not necessarily close to the point sources. In addition, the model resolution can not capture complex orography, meaning that transport of $SO_2$ may not be well

represented. This could be problematic in mountainous areas of USA–W (VanCuren and Gustin, 2015). High resolution model studies would be beneficial to address the both importance of orography and to investigate the $SO_2$ losses close to sources. We also suggest that evaluating UKESM1 against chemical re-analysis fields of $SO_2$ could reduce some of the bias that occurs with using point observations, but there are uncertainties associated with this approach too (Ukhov et al., 2020).

## 6 Conclusions

We evaluate UKESM1 against ground-based and satellite observations of selected sulphur species over the recent historical period. We find that UKESM1 is able to capture the temporal and spatial patterns in surface $SO_2$, surface sulphate concentration, $SO_2$ dry deposition and $TCSO_2$. However, compared to observations we find that the model is biased, depending on the variable, region and species. We address one possible source of bias by introducing a more realistic treatment of $SO_2$ dry
deposition, a key loss process for this species. This change reduces model bias in surface $SO_2$ concentrations and $TCSO_2$. However, it is apparent that other biases exist within the complex sulphur cycle and we highlight some key areas for further investigation and development.

Our evaluation suggests uncertainty in UKESM1's sulphur chemistry is also an important driver of the biases seen in this
study, particularly over polluted regions. Two priorities for further investigation into the oxidation of $SO_2$ in UKESM1 are (i) an evaluation of the CRI-Strat scheme and (ii) a more realistic treatment of cloud water pH. The model's necessarily limited DMS chemistry may also contribute to bias in atmospheric sulphur loading over remote ocean areas. The impact of the nitrate scheme currently in development for UKESM1 will also be investigated in relation to the sulphur cycle. Another aspect of UKESM1's sulphur cycle that would benefit from more detailed analysis is the ratio between wet and dry deposition and how
this compares with observations. We also suggest that high resolution studies to investigate $SO_2$ deposition close to sources would be beneficial for a better understanding of these processes. Finally, we suggest that the $SO_2$ emissions may be too high through a possible combination of too high emissions in the CMIP6 inventory and injection of the emissions into the surface layer only.

The sulphur cycle is a key area of analysis and development for UKESM1 given its importance as a driver of historical aerosol forcing. UKESM1 is relatively unique amongst models in CMIP6 in that it has a fully interactive atmospheric chemistry scheme coupled to a two-moment (mass and number) aerosol scheme. Given the complexity of the model's chemistry-aerosol treatment within the ES framework, the model's performance here is encouraging and provides confidence in UKESM1, particularly in regard to capturing the historical trend. However there is always space for improvement and to the more realistic treatment of
$SO_2$ dry deposition will therefore be incorporated into the planned release of UKESM1.1. This latest model version will be documented in an forthcoming publication which will also address the impact of the $SO_2$ dry deposition changes on aerosol

loading and climate.

*Code availability.* The UM is the source code for the atmosphere-land–ocean–sea ice components of the UKESM1 physical model, including the NEMO and CICE modules for oceans and sea ice, respectively. The UM source code base also houses the GLOMAP-Mode and UKCA modules. JULES is the source code for land and terrestrial biogeochemistry components. MEDUSA is the source code for the ocean biogeochemistry. Due to intellectual property rights restrictions, we cannot provide either the source code or documentation papers for the UM or JULES. *Obtaining the UM.* The Met Office Unified Model is available for use under licence. A number of research organisations and national meteorological services use the UM in collaboration with the Met Office to undertake basic atmospheric process research, produce forecasts, develop the UM code and build and evaluate Earth system models. For further information on how to apply for a licence, see http://www.metoffice.gov.uk/research/modelling-systems/unified-model (last access: 17 March 2021). *Obtaining JULES.* JULES is available under licence, free of charge. For further information on how to gain permission to use JULES for research purposes, see http://jules-lsm.github.io/access_req/JULE_access.html (last access: 17 March 2021). Information about the UKESM1 release and its components and the prerequisites for using it can be found here: http://cms.ncas.ac.uk/wiki/UM/Configurations/UKESM. Briefly, UKESM1 is distributed and run as a Rose suite on the Archer2 and Monsoon computing platforms administered by UK Research Innovation (UKRI) and the Met Office/Natural Environment Research Council (NERC), respectively. Rose is a framework for developing and running meteorological applications and is described in more detail here: http://cms.ncas.ac.uk/wiki/RoseCylc.

*Data availability.* The simulation data used in this study are archived on the Earth System Grid Federation (ESGF) node (https://esgf-node.llnl.gov/projects/cmip6/, last access: 17 March 2021). The model source ID is UKESM1-0-LL for UKESM1. UKESM1 historical simulations are identified by the following variant labels: r1i1p1f2, r2i1p1f2, r8i1p1f2 and r9i1p1f2, (https://doi.org/10.22033/ESGF/CMIP6.6113; Tang (2019)). We acknowledge the use of the CASTNet data base (https://www.epa.gov/castnet, last access: 17 March 2021). Information on the EMEP network can be found in Tørseth et al. (2012) and the data is available from http://ebas.nilu.no/. OMI total column SO2 data was obtained from NASA's Goddard Earth Sciences Data and Information Services Center (GES DISC, https://disc.gsfc.nasa.gov/, last access: 17 March 2021).

## Appendix A

Here we describe in detail the changes to UKESM1's parameterization of dry deposition of $SO_2$ to the surface. These modifications account for the impact of surface wetness due to rainfall or humidity on $SO_2$ dry deposition to vegetated or soil surfaces. We first derive a parameter, *zwet* that designates whether a model grid box at time step $N$ is wet or dry. We assume that on entering time step $N$ the grid box in question has a dry surface (*zwet* = 0). If, during time step $N$, precipitation is greater than a threshold value (set to $0.5\,\mathrm{mm\,day^{-1}}$) the grid box then becomes classed as wet (*zwet* = 1). Once precipitation stops, the grid box is assumed to dry out over a specified period. Assuming no precipitation falls during this drying period, at the end of the period the grid box will be classed as dry (*zwet* = 0). If, during the drying period, a new precipitation event occurs at

the grid box, of intensity greater than $0.5\,\mathrm{mm\,day^{-1}}$, *zwet* is reset to 1. If the precipitation event is less than $0.5\,\mathrm{mm\,day^{-1}}$ but greater than $0.0\,\mathrm{mm\,day^{-1}}$, *zwet* is not reset to 1 and neither is it decreased in value (i.e. no drying is assumed to occur over

that time step). A grid box is classed as wet whenever *zwet* > 0. We tested a range of time periods from three hours to one day and found only minor sensitivity to this parameter. For UKESM1-SO2 we used a drying period of three hours. In future work we will investigate making this parameter a function of surface evaporation and downwelling solar radiation.

If a grid box is classed as wet then $R_{soil}$ and $R_{cut}$ (for all vegetation types) is set equal to $1\,\mathrm{s\,m^{-1}}$. Through Equation 5, $R_c$

will then tend towards a value of $1\,\mathrm{s\,m^{-1}}$, equating to minimal resistance to $SO_2$ deposition. In these situations, the amount of $SO_2$ deposited will primarily be limited by the efficiency of gas transport to the receptor surface, i.e. by $R_a$ and $R_b$ in Equation. 2. If the grid box is classed as dry, $R_{cut}$ is calculated following Equation 9 in Erisman et al. (1994), whereby $R_{cut}$ is a decreasing function of near surface relative humidity. If a grid box surface temperature lies between -1°C and -5°C, $R_{cut}$ is reset to $200\,\mathrm{s\,m^{-1}}$ and below -5°C to $500\,\mathrm{s\,m^{-1}}$, irrespective of the near surface relative humidity. For dry grid boxes, $R_{soil}$

uses a value of $213.5\,\mathrm{s\,m^{-1}}$ for all surface temperatures (see Table A2). For dry surfaces, $R_{cut}$ also approaches a value of $1\,\mathrm{s\,m^{-1}}$ as near surface relative humidity approaches a value of one. As for a wet surface, in these conditions there is minimal resistance to $SO_2$ deposition and $R_c$ will tend towards $1\,\mathrm{s\,m^{-1}}$. Thus $SO_2$ dry deposition will primarily be limited by $R_a$ and $R_b$.

**Table A1.** Summary of the representation of $R_{cut}$ and $R_{soil}$ in UKESM1 and UKESM1-SO2. Units for resistance values are $\mathrm{s\,m^{-1}}$

| Grid box condition | Surface temperature | Near surface relative humidity (RH) | UKESM1 $R_{soil}$ | $R_{cut}$ | UKESM1-SO2 $R_{soil}$ | $R_{cut}$ |
|---|---|---|---|---|---|---|
| Wet | N/A | N/A | 213.5 | $R_{surf}{}^{*}$ | 1.0 | 1.0 |
| Dry | > -1°C | RH < 0.813 | 213.5 | $R_{surf}$ | 213.5 | $R_{cut}{=}2.5\times10^{-4}$ exp[-6.93*RH] ** |
| Dry | > -1°C | RH > 0.813 | 213.5 | $R_{surf}$ | 213.5 | $R_{cut}{=}0.58\times10^{12}$ exp[-27.8*RH] ** |
| Dry | > -5°C and < -1°C | N/A | 213.5 | $R_{surf}$ | 213.5 | 200.0*** |
| Dry | < -5°C | N/A | 213.5 | $R_{surf}$ | 213.5 | 500.0$^{(***)}$ |

* See values in Table A2; ** Following Erisman et al. (1994); *** Irrespective of near surface relative humidity

**Table A2.** Standard surface resistance ($R_{surf}$) values for $SO_2$ for land use types in UKESM1. These values were calculated based on the data published in Zhang et al. (2003) for $SO_2$.

| Land surface type | $R_{surf}$ / s m$^{-1}$ |
|---|---|
| Broad leaf deciduous | 137.0 |
| Broad leaf evergreen tropical | 111.1 |
| Broad leaf evergreen temperate | 111.9 |
| Needle leaf deciduous | 131.3 |
| Needle leaf evergreen | 130.4 |
| C3 grass | 209.8 |
| C3 crop | 30.0 |
| C3 pasture | 209.8 |
| C4 grass | 196.1 |
| C4 crop | 30.0 |
| C4 pasture | 196.1 |
| Shrub deciduous | 185.8 |
| Shrub evergreen | 196.1 |
| Urban | 180.7 |
| Water | 1.0 |
| Soil | 213.5 |
| Ice | 215.1 |

## Appendix B

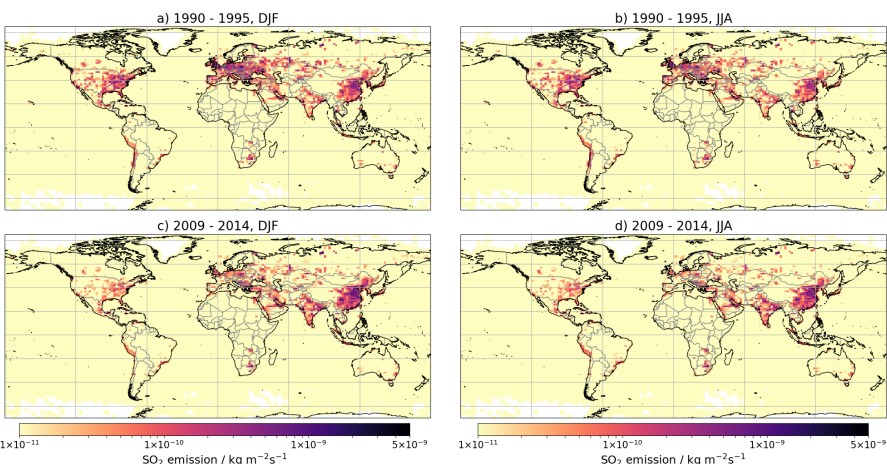

**Figure B1.** Global emissions of $SO_2$ used in UKESM1. Mean seasonal emissions are shown for the 1990–1995 time slice for DJF (a) and JJA (b), and the 2009–2014 time slice for DJF (c) and JJA (d).

## Appendix C

**Table C1.** Global $SO_2$ budget for UKESM1 and UKESM1-SO2. Units for production and loss fluxes are in Tg $[S]\,yr^{-1}$, burdens are in Tg and lifetimes are in days. The values are calculated from a 2 year AMIP simulation covering the period 1981–1983 inclusive.

| | UKESM1 | UKESM1-SO2 |
|---|---|---|
| **Emission sources** | | |
| Surface emission | 61.03 | 61.03 |
| High-level emission | 0 | 0 |
| Natural emission | 14.02 | 14.02 |
| **Sources from DMS oxidation** | | |
| $DMS + OH \rightarrow SO_2$ | 6.42 | 6.37 |
| $DMS + OH \rightarrow SO_2 + MSA$ | 6.36 | - |
| $DMS + OH \rightarrow 0.6SO_2 + 0.4DMSO$ | - | 3.83 |
| $DMS + NO_3 \rightarrow SO_2$ | 3.48 | 3.47 |
| $DMSO + OH \rightarrow 0.6SO_2$ | - | 1.53 |
| $DMS + O(^3P) \rightarrow SO_2$ | 0.1708 | 0.1738 |
| **Sources from COS oxidation** | | |
| $COS + O(^3P) \rightarrow CO + SO_2$ | $7.92 \times 10^{-3}$ | $7.87 \times 10^{-3}$ |
| $COS + OH \rightarrow CO_2 + SO_2$ | 0.105 | 0.106 |
| $COS + hv \rightarrow CO + SO_2$ | 0.024 | 0.0235 |
| **Losses from gas-phase oxidation** | | |
| $SO_2 + OH \rightarrow SO_3{}^{2-} + HO_2$ | 18.66 | 14.86 |
| $SO_2 + O_3 \rightarrow SO_3{}^{2-}$ | $8.19 \times 10^{-4}$ | $6.54 \times 10^{-4}$ |
| $SO_3{}^{2-} + H_2O \rightarrow H_2SO_4 + H_2O$ | - | - |
| $SO_3{}^{2-} + hv \rightarrow SO_2 + O(^3P)$ | $4.07 \times 10^{-9}$ | $4.46 \times 10^{-9}$ |
| **Losses from aqueous-phase oxidation** | | |
| $HSO_3{}^- + H_2O_2 \rightarrow SO_4{}^{2-}$ | 20.52 | 16.46 |
| $HSO_3{}^- + O_3 \rightarrow SO_4{}^{2-}$ | 0.2372 | 0.1625 |
| $SO_3{}^{2-} + O_3 \rightarrow SO_4{}^{2-}$ | 9.22 | 6.35 |
| **Dry Deposition** | 29.49 | 42.56 |
| **Wet Deposition** | 14.09 | 10.57 |
| **Total Sources** | 91.62 | 90.56 |
| **Total Losses** | 92.22 | 90.96 |
| **Burden** | 0.54 | 0.41 |
| **Lifetime** | 2.11 | 1.62 |
| **Oxidation lifetime** | 3.997 | 3.901 |
| **Dry deposition lifetime** | 6.59 | 3.47 |
| **Wet deposition lifetime** | 13.80 | 13.96 |

*Author contributions.* CGJ, JPM, STT and CH contributed to the conceptualization of this study CGJ and STR developed and tested the code changes for the revised dry deposition parameterization. STR ran the UKESM1 and UKESM1-SO2 historical simulations. CJ ran the UKESM1 and UKESM1-SO2 AMIP simulations and calculated the $SO_2$ budgets. CH led the analysis and did the data visualization. RP with support from CL evaluated UKESM1 against the OMI $SO_2$ product. CH led the preparation of the manuscript, with important text contributions from JPM, CGJ and RP. All co-authors contributed invaluable comments in reviewing and editing the manuscript.

*Competing interests.*

The authors declare that they have no conflict of interest.

*Acknowledgements.* For making their measurement data available to be used in this study, we would like to acknowledge the EMEP and CASTNet measurement networks along with any data managers involved in data collection. The EBAS database has largely been funded by the UN-ECE CLRTAP (EMEP), AMAP and through NILU internal resources. Specific developments have been possible due to projects like EUSAAR (EU-FP5)(EBAS web interface), EBAS-Online (Norwegian Research Council INFRA) (upgrading of database platform) and HTAP (European Commission DG-ENV)(import and export routines to build a secondary repository in support of www.htap.org). The US Environmental Protection Agency is the primary funding source for CASTNet, with contracting and research support from the National Park Service. This study used JASMIN, the UK's collaborative data analysis environment (http://jasmin.ac.uk) (Lawrence et al., 2013). CH made extensive use of the Iris and Cartopy libraries for the analysis and data visualisation in this study (Met Office, 2010 - 2013, -).

*Financial support.*

This research has been supported by the UK Government, BEIS and DEFRA (grant no. GA01101), the National Environmental Research Council (NERC) National Capability Science Multi-Centre (NCSMC) (UKESM (grant no. NE/N017978/1)). CJ acknowledges funding from the EU Horizon 2020 project CRESCENDO, grant number: 641816. RJP was supported by the UK Natural Environment Research Council (NERC) by providing funding for the National Centre for Earth Observation (NCEO) through grant number NE/R016518/1. STT has been supported by the UK–China Research and Innovation Partnership Fund through the Met Office Climate Science for Service Partnership (CSSP) China as part of the Newton Fund. OMI $SO_2$ product development and analysis have been supported by NASA Earth Science Division (ESD) Aura Science Team program (Grant #80NSSC17K0240).

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
