# Peer review of "Evaluation of $SO_2$ , $SO_4^{2-}$ and an updated $SO_2$ dry deposition parameterization in UKESM1"

_Atmospheric Chemistry and Physics, 2021_

## Author Response (AR1)

**Reply to Reviewer 1**

We thank the reviewer for their careful reading of the manuscript and their suggestions for improvement. The Reviewer's comments and our responses can be found below.

**[Specific comments]**

1. The crux of this study should be the impact of changing the dry deposition parameterization on the simulated tropospheric sulfur cycles. In order to serve for this purpose, the authors should beef up the contents of the manuscript in the model description and the discussion of dry deposition velocities. In section 2.2.1, the authors should provide a clear description of mathematical formulae employed for prescribing the leaf cuticle and soil resistances to SO2 uptake as a function of relative humidity and references for the basis of the employed formulae. Section 4 should begin with the discussion of simulated dry deposition velocities themselves (rather than the dry deposition fluxes) before and after the implementation of the new parameterization (see, for example, Ganzeveld et al., 1998).

=> We have now included an Appendix (Appendix A) to provide a more detailed description of the changes to the SO2 dry deposition parameterization. The reader is directed to this from Section 2.2.1. This appendix includes a table illustrating how cuticular restistance (Rcut) and soil resistance (Rsoil) in are represented differently in UKESM1 and UKESM1-SO2. We include the mathematical formulae for calculating Rcut as a function of humidity and clearly state the references used for these formulae. In keeping the technical details of the SO2 dry deposition modifications separate from the main text we aim to keep the manuscript more streamlined for the reader.

=> As suggested by the reviewer we have now included an analysis of the SO2 dry deposition velocities in UKESM1 and UKESM1-SO2. In section 4.1 we have included a new figure showing the annual mean, DJF mean and JJA mean for SO2 dry deposition velocities for UKESM1, and difference plots showing the impact of the modifications to the SO2 dry deposition paramterization on the deposition velocities in UKESM1-SO2. These results are then summarized in Section 4.1, paragragh 1.

=> In response the comment in paragraph 1 of the reviewer's General Comments, we clarify here that we do not mean to suggest that we have developed a new treatment of surface canopy wetness. Rather that this is an update for the SO2 dry deposition parameterization in UKESM1. We have reworded the text in Section 2.1.1 to clarify that we have adapted the findings from observational studies and applied them to UKESM1.

2. It is not clear whether other aerosol-climate and earth system models share some of the model biases reported in this study, which I hope the authors will touch on when revising the manuscript. The authors allude to inaccuracies in CMIP6 emissions as one of the error sources. It leads me to wonder if other models participating in CMIP6 exhibit the same problem as identified in this study. In addition, the authors need to elaborate the point of argument by Pope and Chipperfield (2021) regarding "total SO2 emissions in CMIP6 are moderately larger than the HTAP-OMI and EDGAR data sets" (L576-579).

=> With the exception of Aas et al., (2019) we are not currently aware of a study that has evaluated SO2/SO4(2-) across the CMIP6 ensemble. Aas et al. (2019) compare modelled and observed trends in SO2 and SO4 over the recent historical period, finding that the

models do capture these trends over most regions. This is also the case for UKESM1 and UKESM1-SO2. We have added a snetence in Section 3.1, Para. 1 to highlight the agreement between out findings and those of Aas et al., (2019). However, it is not clear from Aas et al. (2019) how the models are are biased relative to SO2 or SO4(2-) concentrations. We note that in a communication from Stephen Smith in the open discussion of this manuscript that the CMIP6 emissions over the Western USA are too high, which we have highlighted in our revised manuscript, along with an elaboration on the statement regarding the differences between the CMIP6 dataset and the HTAP-OMI and EDGAR data sets (see Discussion, Para. 3). Further, Aas et al., (2019) also suggest that there are larger uncertainties in the emissions and representivity of the SO2 emissions over East Asia compared with Europe and North America, although the emission datasets used in that study were different from the CMIP6 emissions used in this study. We aim to evalaute sulphur species in the wider CMIP6 SO2 emissions dataset.

3. The analysis of the present model results will become much stronger if the authors can dive deeper into the metrics of model behavior related to the budget of atmospheric sulfur compounds and its changes with the revision of the dry deposition parameterization. This will allow us to grasp the broader context of this study. For example, the authors could calculate the regional lower-tropospheric budgets of SO2 and sulfate following Chin and Jacob (1996, Figs. 2-3) or re-iterate the global budget of SO2 discussed in Mulcahy et al. (2020, Tables 4-5) with possible extension of the comparison with yet other models. The point is that knowing the proportions of SO2 lost via dry and wet deposition and via oxidative conversion to sulfate provides a more in-depth measure of UKESM1's performance in its sulfur cycle. As it stands in the present version of the manuscript, this aspect is discussed only qualitatively. Another useful metric would be the SO2 lifetime and its seasonal variations estimated from regional SO2 vertical column densities and emission intensities (e.g., Lee et al., 2011, Fig. 2; Buchard et al., 2014, Fig. 3).

=> We have now calculated the SO2 budget using a 2 year AMIP simulation covering the period 1981–1983 inclusive. The SO2 budget for UKESM1 is in good agreement with that shown in Mulcahy et al. (2020). The budget for UKESM1-SO2 clearly shows the impact of the dry deposition modifications on the SO2 burden, SO2 lifetime and the deposition and oxidation processes. The SO2 budget is presented in Appendix C, with the main findings summarized in Section 4.1, para. 2 and discussed in the Discussion, para. 5 and 7.

4. The authors state that emissions from the energy and industrial sectors are all emitted into the first model layer (line 156), which seems to have been indicated by Mulcahy et al. (2020) as one potential weakness for the handling of this process in UKESM1. The injection of SO2 emissions from large stacks across several vertical layers above the lowest model layer (in lieu of plume-rise modeling) has improved the agreement of ground-level SO2 concentrations simulated by GEOS-5/GOCART model with observations in USA, whereas the SO2 vertical column densities did not change significantly (Buchard et al., 2014). The authors should refer to this finding when discussing the model evaluation against observed ground-level SO2 concentrations. Perhaps it is too much to ask a new set of model runs for testing this emission treatment problem within the present study, but I am inclined to an idea that it can alleviate many problems identified in this study (high biases in the ground-level aerosol sulfate concentrations in USA and Europe).

=> The reviewer is correct in that varying the height at which SO2 emitted in to the atmosphere, as opposed to emitting all SO2 at the surface, can reduce model bias in surface SO2 concentrations. This was demonstrated by Mulcahy et al. (2020) in their comparison between UKESM1 and HadGEM-GC3.1, although model bias in atmospheric SO4 concentrations was not reduced in this study. In emitting all SO2 at the surface in UKESM1 we maintain consistency with other Met Office model configurations, however, implementing a varying emission height is a key development target for the model. We thank the reviewer for pointing us towards the study by Buchard et al. (2014) and we have cited this paper in the Discussion, para. 4.

5. The reactive uptake of SO2 on dust aerosols can notably reduce the SO2 concentrations and has a very large impact over China (e.g., Dentener et al., 1996; Liao et al., 2003, Bauer and Koch, 2005). It doesn't appear that UKESM1 accounts for this process, hence another possible contributor to the model SO2 bias especially over China. The authors should justify the change of reference height from 50 m to 10 m for the computation of aerodynamic resistance, by explaining whether it comes with changes in the configuration of vertical layer thickness of the model. The reference height should be in general approximately half the thickness of the lowest model layer (e.g., Ganzeveld and Lelieveld, 1995, Section 3.2); if the lowest model layer thickness is substantially greater than 20 m (say, 40 m or greater), it calls for a strong rationale for using the reference height at 10 m. The authors should clarify the point of argument by Holtslag and De Bruin (1988) if UKESM1's lowest model layer thickness is much greater than 20 m. Toyota et al. (2016, Section 2.2) gave a rationale in favor of Ganzeveld and Lelieveld (1995) for the choice of reference height from the mathematical formulation of aerodynamic resistance. Toyota et al. (2016) also noted that stability corrections applied for the computation of aerodynamic resistance are often inconsistent between dry deposition and host meteorological modules employed in the same model system. Does the use of the Holtslag and De Bruin (1988) function instead of the Dyer (1974) function reduce or eliminate this problem of inconsistency with meteorological flux calculation (i.e., u\* and L) within UKESM1?

=> The reviewer is correct in that UKESM1 does not account for the reactive uptake of dust on aerosols and we thank the reviewer for highlighting this potential source of bias in the model. Although we do not have specific plans to include this process in the model, it could be a target for future development.

=> We updated to Holtslag and De Bruin (1988) because it is considered a slightly more up-to-date set of equations for describing the fluxes in the boundary layer. In addition, by changing the the reference height from 50 m to 10 m we are making it more more consistent with the height of the lowest model level, which is 20 m in UKESM1. Unfortunately using Holtslag and De Bruin (1988) instead of Dyer does not increase consistency between meteorological stability functions and those used in UKESM1/UKCA. We are interested in updating the meteorological fluxes to also use Holtslag and De Bruin (1988) but this was not possible within this study. We agree with the reviewer that ideally there would consistency across the various flux formulations and we will aim study this, though changing the flux formulation across the meteorological subroutines is a major task.

**[Technical suggestions]**

1. L158-159: "Gas- and aqueous-phase oxidation of ..."

=> Corrected as suggested

2. L113: Would you classify the gravitational settling as part of the wet deposition processes?

=> The approach for dry deposition of aerosol in GLOMAP-mode within UKCA (where UKCA is the chemistry and aerosol model in UKESM) is the same as that described in Section 2.2.2 of Mann et al.(2010) with a dry deposition velocity (Vd) for each aerosol mode given as the combination of a gravitational settling velocity (Vgrav) and one-over the sum of the aerodynamic and surface resistances (Ra and Rs) i.e. Vd = Vgrav + (1 / Ra + Rs).

(Mann et al., Description and evaluation of GLOMAP-mode: a modal global aerosol microphysics model for the UKCA composition-climate model, Geosci. Model Dev., 3, 519–551, https://doi.org/10.5194/gmd-3-519-2010, 2010.)

3. L126: "ares" -> "areas"

=> Thank you for highlighting theis error. Corrected.

4. L136: "to be developed" -> "being developed"

=> Sentence modified to "UKESM1 is the latest generation Earth System (ES) model developed in the UK." UKESM1 is the first generation of the model to be developed and was released in Jan 2020. As such it is not quite correct to say that this model version is still being developed.

5. L190: The soil pH is not taken care of in the model even after the revision to the parameterization, right? Please clarify. The authors may also want to cite Ganzeveld et al. (1998), which dealt with changes in the soil pH in their global tropospheric sulfur chemistry-transport model.

=> The reviewer is correct, soil pH is not accounted for in UKESM's SO2 dry deposition parameterization.

6. L201: Garland and Branson (1977) reported the dry deposition of SO2 to pine forest. Please correct me if I am wrong, but I cannot find the surface resistance of SO2 on the water surface in this study.

=> Thank for highlighting this error. We have now used the correct citation i.e., Garland, J. A., The dry deposition of sulphur dioxide to land and water surfaces, Proc. R. Soc. Lond. A. 354, 245-268 (1977)

7. Figure 2 caption: "(a, c)" -> "(a, d)", "(b, d)" -> "(b, e)" and "(c, e)" -> "(c, f)"

=> Thank you for highlighting theis error.Corrected.

8. L376, 485 & 488: "peninsular" -> "peninsula"

=> Thank you for highlighting theis error. Corrected.

9. Table 6 caption: Please come up with a better phrase for what "zonally averaged (median) time series" mean.

=> We have re-worded this caption to clearly describe what the data represents.

10. Figure 9: Change the figure title for "South East Asia" to "South to North East Asia".

=> This has now been corrected

11. L486: "9" -> "8"

=> Thank you for highlighting theis error. Corrected.

12. L503: "this aspect of THE change"

=> Thank you for highlighting theis error. Corrected.

13. L538: "NMB = 0.25" -> "NMB = -0.25" and "NMB = 0.43" -> "NMB = -0.43"

=> Thank you for highlighting theis error. Corrected.

14. L579: The authors need to be more specific about the data merging between OMI and HTAP.

=> We have now included a reference to the relevant citation Liu et al., (2018) in Discussion, para. 3 as we didn't create the data set, just explore it. If the reviewer or reader are interested the details of how the data set was derived, we suggest they follow the reference.

15. L690: "would BE beneficial"

=> Thank you for highlighting theis error. Corrected.

**[References]**

Bauer, S. E., and D. Koch (2005), Impact of heterogeneous sulfate formation at mineral dust surfaces on aerosol loads and radiative forcing in the Goddard Institute for Space Studies general circulation model, J. Geophys. Res., 110, D17202, doi:10.1029/2005JD005870.

Buchard, V., da Silva, A. M., Colarco, P., Krotkov, N., Dickerson, R. R., Stehr, J. W., Mount, G., Spinei, E., Arkinson, H. L., and He, H.: Evaluation of GEOS-5 sulfur dioxide simulations during the Frostburg, MD 2010 field campaign, Atmos. Chem. Phys., 14, 1929–1941, https://doi.org/10.5194/acp-14-1929-2014, 2014.

Chin, M., and D. J. Jacob (1996), Anthropogenic and Natural Contributions to Tropospheric Sulfate: A Global Model Analysis, J. Geophys. Res., 101 (D13), 18691, doi:10.1029/96jd01222.

Dentener, F. J., G. R. Carmichael, Y. Zhang, J. Lelieveld, and P. J. Crutzen, Role of mineral aerosol as a reactive surface in the global troposphere, J. Geophys. Res., 101, 22,869-22,889, 1996.

Dyer, A.: A review of flux-profile relationships, Boundary-Layer Meteorology, 7, 363-372, 1974.

Galmarini, S., Makar, P., Clifton, O., Hogrefe, C., Bash, J., Bianconi, R., Bellasio, R., Bieser, J., Butler, T., Ducker, J., Flemming, J., Hozdic, A., Holmes, C., Kioutsioukis, I., Kranenburg, R., Lupascu, A., Perez-Camanyo, J. L., Pleim, J., Ryu, Y.-H., San Jose, R., Schwede, D., Silva, S., Garcia Vivanco, M., and Wolke, R.: Technical Note – AQMEII4 Activity 1: Evaluation of Wet and Dry Deposition Schemes as an Integral Part of Regional-Scale Air Quality Models, Atmos. Chem. Phys. Discuss., https://doi.org/10.5194/acp-2021-313, in review, 2021.

Ganzeveld, L., and Lelieveld, J. (1995), Dry deposition parameterization in a chemistry general circulation model and its influence on the distribution of reactive trace gases, J. Geophys. Res., 100(D10), 20999- 21012, doi:10.1029/95JD02266.

Ganzeveld, L., Lelieveld, J., and Roelofs, G.-J. (1998), A dry deposition parameterization for sulfur oxides in a chemistry and general circulation model, J. Geophys. Res., 103(D5), 5679-5694, doi:10.1029/97JD03077.

Hayden, K., Li, S.-M., Makar, P., Liggio, J., Moussa, S. G., Akingunola, A., McLaren, R., Staebler, R. M., Darlington, A., O'Brien, J., Zhang, J., Wolde, M., and Zhang, L.: New methodology shows short atmospheric lifetimes of oxidized sulfur and nitrogen due to dry deposition, Atmos. Chem. Phys., 21, 8377–8392, https://doi.org/10.5194/acp-21-8377-2021, 2021.

Holtslag, A. A. M., and De Bruin, H. A. R.: Applied Modeling of the Nighttime Surface Energy Balance over Land. J. Appl. Meteorol., 27, 689-704, 1988.

Lee, C., Martin, R. V., van Donkelaar, A., Lee, H., Dickerson, R. R., Hains, J. C., Krotkov, N., Richter, A., Vinnikov, K., and Schwab, J. J.: SO2 emissions and lifetimes: Estimates from inverse modeling using in situ and global, space-based (SCIAMACHY and OMI) observations, J. Geophys. Res., 116, D06304, doi:10.1029/2010JD014758, 2011.

Liao, H., P. J. Adams, S. H. Chung, J. H. Seinfeld, L. J. Mickley, and D. J. Jacob, Interactions between tropospheric chemistry and aerosols in a unified general circulation model, J. Geophys. Res., 108(D1), 4001, doi:10.1029/2001JD001260, 2003.

Mulcahy, J. P., Johnson, C., Jones, C. G., Povey, A. C., Scott, C. E., Sellar, A., Turnock, S. T., Woodhouse, M. T., Abraham, N. L., Andrews, M. B., Bellouin, N., Browse, J., Carslaw, K. S., Dalvi, M., Folberth, G. A., Glover, M., Grosvenor, D. P., Hardacre, C., Hill, R., Johnson, B., Jones, A., Kipling, Z., Mann, G., Mollard, J., O'Connor, F. M., Palmiéri, J., Reddington, C., Rumbold, S. T., Richardson, M., Schutgens, N. A. J., Stier, P., Stringer, M., Tang, Y., Walton, J., Woodward, S., and Yool, A.: Description and evaluation of aerosol in UKESM1 and HadGEM3-GC3.1 CMIP6 historical simulations, Geosci. Model Dev., 13, 6383–6423, https://doi.org/10.5194/gmd-13-6383-2020, 2020.

Pleim, J., and Ran, L.: Surface Flux Modeling for Air Quality Applications, Atmosphere, 2, 271-302, https://doi.org/10.3390/atmos2030271, 2011.

Toyota, K., Dastoor, A. P., and Ryzhkov, A.: Parameterization of gaseous dry deposition in atmospheric chemistry models: Sensitivity to aerodynamic resistance formulations under statically stable conditions, Atmos. Environ., 147, 409-422, 2016.

**Reply to Reviewer 2**

We thank the reviewer for their careful reading of the manuscript and their suggestions for improvement. The Reviewer's comments and our responses can be found below.

**[Specific Comments]**

I am a bit skeptical when it comes to the selection of observations to be used for evaluating trends. More details of this and other specific comments in the bullet points:

1. Line 35-38. Note that the unit is TgSOx (as SO2)

=> Thank you for highlighting theis error. Now corrected.

2. Line 69. "Relatively short lifetime". Relative to what? A lot of species has shorter lifetime than this.

=> We have ammended this statement, please see our response to Comment 3.

3. Line 70. In this general statements it seems like the removal of SO2 is mainly described as deposition of the component. Even though it is described later in the introduction one should mention that oxidation of SO2 to SO4 is important factor for the lifetime of SO2, and that rate is dependent on the oxidation capacity and the acidity of the cloud droplets (which other atmospheric components like NH3 will influence). Maybe here just make it shorter and state that the lifetime of so2 depends on both wet and dry deposition of the molecule and the oxidation rate to SO4?

=> We have reworded this paragraph in line with the reviewer's suggestions. The paragraph is now as follows: '... The lifetime of SO2 depends on both wet and dry deposition of the molecule and the oxidation rate to SO4. The 2 day lifetime is such that much of the loss via oxidation and deposition occurs locally. SO2 loss near sources and the impact of environmental conditions on loss processes have been investigated in a number of studies.'

4. Line 75-90. In this section, the references are really old.

=> We agree that there references in this section are relatively old, these studies having been done from ~1970-1990 when acid deposition over Europe and North America was a serious problem. While there has been a number of measurement studies measuring SO2 dry deposition, particularly the impact of surface wetness on deposition velocity (e.g. Fowler et al., 1995; Feliciano et al., 2001; Matsuda et al., 2006; Tsai et al., 2011; Myles et al., 2012) there appears to have been less drive for the type of studies we reference here i.e. looking at loss rates close to sources. There has also been considerable work to develop parameterizations for SO2 dry deposition in regional and global scale models over the last 20 years or so (e.g. Zhang et al., 2003; Ganzeveld and Lelieveld, 1995; Ganzeveld et al., 1998). We particularly cite Garland and Branson, 1977; Fowler et al., 1978; Erisman and Baldocchi, 1994 and Erisman et al., 1994 as they highlight that the loss processes are occuring sub-grid scale compared to UKESM1.

5. Line 119 – 122 starting with "Following the increasingly.." seems a bit oddly placed in the introduction. Should maybe be moved to the beginning as the reason why monitoring are being conducted?

=> We have re-worded this sentence as follows: 'In the 1970's and 1980's the increasingly detrimental impacts of rising SO2 emissions on acid deposition, air quality and human health in Europe and North America led to monitoring networks being set up in these regions...'

6. Line 118. I don't really agree with the statement of "the main challenge to capture historical trends" is the lack of observations. Sulfur is one of the species that has been monitored the most in especially Europe and North America. But there are of course regions in the world where this stamen is very valid. I assume one of the challenges is the non-linearity in trends, i.e. the dependence on atmospheric chemistry on the sulfur trends, and the lack of a range of data to detailed process studies on a large scale as well as long term flux measurements and not only atmospheric concentrations?

=> We agree with the reviewer in that sulphur species are relatively well observed compared to many atmospheric constituents, at least in Europe and North America for the period from the 1970's/80's to the present day. However, the point we would like to make is that even with these data sets we can still only evaluate part of the model's historical simulation (which runs from 1850 - 2014), and similar data sets are not available for other major source regions such as India, China and the Middle East, or remote regions. While the satellite observations of SO2 can help with spatial coverage they are somewhat temporally limited (with regard to evaluating the historical period) and capture only SO2 through the column. We do also mean to allude to the lack of data available for large scale

process studies, i.e. flux measurements and co-located measurements for relevant species. We have ammended the paragraph to capture these points as follows: 'Sulphur species are relatively well observed compared to many atmospheric components as their role in air pollution is well established. In the 1970's and 1980's the increasingly detrimental impacts of rising SO2 emissions on acid deposition, air quality and human health in Europe and North America led to monitoring networks being set up in these regions (Torseth 2012; CASTNET 2004). Rising pollution in Asia also led to the establishment of the The Acid Deposition Monitoring Network in East Asia (EANET) in 2001 (e.g. Wang 2008). However, even with these data sets it is only possible to evaluate model simulations of the recent historical period and similar data sets are not available for other large source regions such as India, the Middle East, or remote regions. Further, the lack of a range of measurements, including flux observations, hinders detailed process studies at large scales.'

7. Chapter 2.4. It seems like it is not a criterion to have co-located SO2 and SO4 observations and then you could have benefited from also using SO4 aerosol data from IMPROVE. Has that been considered since you state in the beginning that too little data is hampering the comparison with models?

=> The reviewer makes a valid point regarding the IMPROVE data set. However a clear advantage of the CASTNet data for this study is the co-location of the SO2 and SO4 observations, and SO2 dry deposition data. While we are not quite doing a full process analysis here, we are trying to understand bias in UKESM1's representation of sulphur cycle, including processes. In addition we seek to improve the model by modifying the dry deposition process and it is valuable to be able to link the impact of those changes through comparisons of dry deposition, SO2 concentration and sulphate. Please note that we have also ammended our EMEP data set, limiting the measurement sites to those which have longer term data sets and both SO2 and SO4 available (see also the response to Comment 9). We have updated Section 2.4 to include this information.

8. Line 251. SO2 and SO4 are measured with filter pack sampler and weekly sampling intervals, not hourly measurements as stated.

=> Thank you for highlighting theis error. Now corrected.

9. The number of sites has varied through the period and it seems like you have used all the sites without considering the length of the time series? If so, have you compared the trends using only sites with observations for the whole period? Especially in Europe the differences in site density throughout the period may influence the trend. In the beginning it was less sites in Southern Europe. Information about the number of sites should be included in the figure (and in table 3).

=> In updating the EMEP data set to include the data from 2010-2014 we also revised the sites we used in the study. We limited the EMEP sites to those which had at least 10 years of continuous measurements (and in general where SO2 and SO4 measurements were co-located, see also the reply to Comment 7). In this study we have only considered the European region as a whole. Figure 6 shows that the model bias is indeed variable across Europe, and similarly to the USA, the biases are larger in the more polluted regions. As part of our wider analysis we did produce the plots shown in Figure 6 for the two time

slices (1990-1995) and 2009-2014). However, the results were not substantially different from those for the whole time period, i.e model bias in SO2 and SO4 concentrations was lower at cleaner sites compared with polluted sites, (see also the response to Comment 12). Therefore we did not see any clear benefit to the manuscript. We have now included the number of sites used to produce Figures 2, 3, 4, 5, 6, 7 and Tables 3, 4, 5, 7 (see also the response to Comment 13).

10. Fig2 and Fig 4 (and fig 11) Why does SO4 in Europe only include data up to 2010? Surely there are observations after 2010 in EMEP, found in http://ebas.nilu.no/ (this database should also be included in the section of data availability). In Figure 6 it seems like you have used data up to 2014?

=> We have now included the full EMEP data set up to 2014 for SO2 and SO4 and the relevant plots and statistical data have been modified (Figures 2, 4, 6, 7, 11 and Tables 3, 4, 5, 7). We have also included the website for the EMEP observations in the data availability section.

11. Fig 3 and Fig 4. For the average annual concentrations for the different 5 years period. Have you used a criteria for the data capture needed to make an average. E.g 75%, 3 out of 5 years etc?

=> Yes, for the 5 year time slice statistics we have only used sites for which there was at least 3 out of 5 years. We have now added this information to Section 2.5.

12. Chapter 3.3. Have you calculated the per cent bias? That may give a different geographical distribution of the bias than absolute concentrations. In addition, it would be interesting to know whether the model is able to capture the per cent changes (trends) at the different sites, that will give further insight if the model and observations are responding similar to the emission changes.

=> As part of our analysis we did look at percentage bias in addition to normalised mean bias (NMB) for the plots in Section 3.3 (Figures 5 and 6). However the percentage bias plots do not look substantially different from the NMB plots. The model bias (absolute or normalised) is greatest at the sites where the surface concentrations are highest. Given that the bias is very large at some sites, we felt that the NMB plots were easier to interpret than the plots of percentage bias, which could be much greater 100% in some cases. We also looked at the bias during the two time slices. Again, these plots did not show substantial difference from the plots for the full time period, i.e. the sites where the surface concentrations of SO2 ansd SO4 were greatest also had the hingest bias. We have chosen not to focus on individual sites in this study as comparisons between point observational data and model grid cell output is difficult to justify in isolation, particularly for SO2 where loss processes (e.g. deposition/oxidation) are occuring within a single grid cell. We have summarised the trends in the modelled and observational data in Tables 3 and 4 where we report the trend for the two time slices and over the full period. While we acknowledge that some sites may be very different, we believe these results are indicative of the model's behaviour in the different regions.

13. Fig 7. I assume the blue shaded areas for the modelled and the black variations in the observations indicate the standard deviations between the sites? Should be mentioned in the caption in addition to how many sites are included in the analysis (also in Table 5).

=> In Figure 7 we have now added a description for what the blue shaded region and the black bars represent. We have also included the number of observational data sets used for each data set in the caption for Figure 7, and in Table 5.

13. Fig 12. In the figure caption you should include the time period you are looking at.

=> We have included the time period (2005-2014) in the caption.

**[References]**

Feliciano et al., Evaluation of SO2 dry deposition over short vegetation in Portugal, Atmospheric Environment, 35, 21, 3633-3643, 2001, https://doi.org/10.1016/S1352-2310(00)00539-2

Environment, 35, 21, 3055-3045, 2001, https://doi.org/10.1010/01352-2510(00)00555-2

Fowler et al., Long term measurements of SO2 dry deposition over vegetation and soil and comparisons with models, Studies in Environmental Science, 64, 9-19, Acid Rain Research: Do we have enough answers?, https://doi.org/10.1016/S0166-1116(06)80269-4,

Matsuda et al., Deposition velocity of O3 and SO2 in the dry and wet season above a tropical forest in northern Thailand, Atmospheric Environment, 40, 39, 7557-7564, 2006, https://doi.org/10.1016/j.atmosenv.2006.07.003

Myles et al., A comparison of observed and parameterized SO2 dry deposition over a grassy clearing in Duke Forest, Atmospheric Environment, 49, 212-218, 2012, https://doi.org/10.1016/j.atmosenv.2011.11.059

Tsai et al., Observation of SO2 dry deposition velocity at a high elevation flux tower over an evergreen broadleaf forest in Central Taiwan, Atmospheric Environment, 44, 8, 1011-1019, 2010, https://doi.org/10.1016/j.atmosenv.2009.12.022

---

## Author Response (AR2)

**Reply to Reviewer 2**

We thank the Reviewer for their careful reading of the revised manuscript. We have clarified the text regarding specific comments 1 and 2, and we have corrected the technical errors as suggested. The Reviewer's comments and our responses are below.

**[Specific Comments]**

1. In response to my question of "Would you classify the gravitational settling as part of the wet deposition processes?", the authors mentioned: "The approach for dry deposition of aerosol in GLOMAP-mode within UKCA (where UKCA is the chemistry and aerosol model in UKESM) is the same as that described in Section 2.2.2 of Mann et al.(2010) with a dry deposition velocity (Vd) for each aerosol mode given as the combination of a gravitational settling velocity (Vgrav) and one-over the sum of the aerodynamic and surface resistances (Ra and Rs) i.e. Vd = Vgrav + (1 / Ra + Rs)." The authors appear to agree that they should classify the gravitational settling as part of the dry deposition process. However, in the revised manuscript (Line 113), the authors still classify the gravitational settling as part of the wet deposition process just as they did in the previous version of the manuscript. I would rephrase "including gravitational settling and rain out" to something like "including nucleation scavenging within the cloud (rainout) and impact scavenging below the cloud (washout)", if it correctly describes the representation of wet deposition in UKESM1.

=> AR: We apologise for neglecting to update the manuscript in line with our response to the Reviewer's original comment. Following the reviewer's suggestions, we have now rectified this in Section 1, para. 7 as follows:

'Deposition of SO4(2-) is mainly via wet processes (approximately 90\%, Chin et al., 2000), including nucleation scavenging within the cloud (rain out) and impact scavenging below the cloud (wash out), but dry deposition of SO4(2-) does occur through gravitational settling.'

2. Paragraph starting from Line 229: After reading Holtslag and de Bruin (1988), I still don't fully agree with the rationale as stated by the authors for lowering the reference height from 50 m to 10 m. Holtslag and de Bruin (1988) used wind speed at 10 m as one of their input parameters while computing the flux-profile relationships up to the height of 80 m. I agree that the Psi function of Holtslag and de Bruin (1988) should work better than that of Dyer (1974) under strongly stable conditions, but I don't necessarily see a point to lower the reference height to 10 m from the argument made by Holtslag and de Bruin. We should note that this change in the reference height also influences the aerodynamic resistance calculation under unstable conditions. As mentioned by the authors in their response to my question, it would be more appropriate to justify the change from the point that the height of the lowest model level is 20 m in UKESM1. It seems also useful to clarify whether UKESM1's dry deposition module takes the value of the Obukhov length (L) directly from the dynamical/physical model component or whether it derives L from other variables computed by the dynamical/physical model component.

=> AR: We thank the reviewer for highlighting several important points regarding the application of the Holtslag and de Bruin (1988) scheme. We clarify that our motivation for dropping the reference height from 50 m to 10 m were (i) it is a more standard approach and (ii) it fits better with the lowest

level of the model and the surface. We have updated the text in Section 2.2.1 (para. 2, L231-235) to better convey our reasons for changing the reference height (z) and to include the reviewer's point that changing z also affects the aerodynamic resistance calculation under unstable conditions.

'Here we update the calculation from that given by Dyer et al. (1974) to that described by Holtslag and de Bruin (1988). We also reduce the reference height for dry deposition (z) from 50 m to 10 m. The reference height is the height below which there is no turbulence in very stable conditions and is also important for calculating Ra. Following Ganzveld and Lelieveld (2005) the reference height should be half the average height of the lowest model layer, which in UKESM1 is 20 m. The changes to z/L and z act to reduce the rate at which the deposition velocity decreases in very stable conditions, although we note that there is also an impact on the calculation of aerodynamic resistance in unstable conditions.'

Regarding the Monin-Obukhov length (L), this is derived locally in the UK Chemistry and Aerosol (UKCA) code using local values of air density, temperature and friction velocity (where the friction velocity is computed in the UM turbulence scheme, so is consistent across subroutines). We have now included this information in Section 2.2.1, para. 2.

**[Technical suggestions]**

=> AR: We thank the Reviewer for their careful reading of the manuscript and for highlighting the typographical errors below. Except for Suggestion 13 (at Line 753), we have corrected the manuscript as suggested.

1. Lines 80 and 81: You might want to remove "2" from "2 200-300 km" and "2 10-15%".

   => Corrected.

2. Table 1 caption: "UKEMS1" -> "UKESM1"

   => Corrected.

3. Line 359: Do you need a negative sign in "-1.31"?

   => Corrected.

4. Line 481: "20-45 W" -> "20–45 N".

   => Corrected.

5. Line 520: Add unit for the dry deposition velocity values.

   => Corrected.

6. Line 523: a factor OF 2-4

   => Corrected.

7. Line 537: "dry deposition burden" -> "dry deposition flux"

   => Corrected.

8. Line 539: "0.54 Tg y-1 and 0.41SO2" -> "0.54 Tg to 0.41 Tg"

   => Corrected.

9. Line 623: larger THAN

   => Corrected.

10. Line 715: IT is unlikely

   => Corrected.

11. Line 716: Add a comma after "sources".

   => Corrected.

12. Line 752: "exits" -> "exist"

   => Corrected.

13. Line 753: Drop "and" between "these" and "target".

We have simplified the sentence from:

'However, it is apparent that other biases exits within the complex sulphur cycle and we highlight some key areas for further investigation to better understand these and target areas for development.'

to:

'However, it is apparent that other biases exist within the complex sulphur cycle and we highlight some key areas for further investigation and development.'

14. Table A1: If the two new formulae for Rcut as functions of RH are both taken from Erisman et al. (1994), they should be both marked with double asterisks.

   => Corrected.

15. Table C1: Correct the notation for bisulfite ($HSO_3^-$), sulfite ($SO_3^{2-}$) and sulfate ($SO_4^{2-}$).

   => Corrected.